# Lnc-mg is a long non-coding RNA that promotes myogenesis

Mu Zhu[1,2,3,*], Jiafan Liu[1,*], Jia Xiao[4,*], Li Yang[1,*], Mingxiang Cai[5], Hongyu Shen[1], Xiaojia Chen[1], Yi Ma[1], Sumin Hu[3], Zuolin Wang[5], An Hong[1], Yingxian Li[2], Yao Sun[5] & Xiaogang Wang[1]

Recent studies indicate important roles for long noncoding RNAs (lncRNAs) as essential regulators of myogenesis and adult skeletal muscle regeneration. However, the specific roles of lncRNAs in myogenic differentiation of adult skeletal muscle stem cells and myogenesis are still largely unknown. Here we identify a lncRNA that is specifically enriched in skeletal muscle (myogenesis-associated lncRNA, in short, lnc-mg). In mice, conditional knockout of lnc-mg in skeletal muscle results in muscle atrophy and the loss of muscular endurance during exercise. Alternatively, skeletal muscle-specific overexpression of lnc-mg promotes muscle hypertrophy. *In vitro* analysis of primary skeletal muscle cells shows that lnc-mg increases gradually during myogenic differentiation and its overexpression improves cell differentiation. Mechanistically, lnc-mg promotes myogenesis, by functioning as a competing endogenous RNA (ceRNA) for microRNA-125b to control protein abundance of insulin-like growth factor 2. These findings identify lnc-mg as a novel noncoding regulator for muscle cell differentiation and skeletal muscle development.

[1] Guangdong Provincial Key Laboratory of Bioengineering Medicine & National Engineering Research Center of Genetic Medicine, Department of Cell Biology and Institute of Biomedicine, Jinan University, Huang-Pu Avenue West 601, Guangzhou 510632, China. [2] State Key Laboratory of Space Medicine Fundamentals and Application, China Astronaut Research and Training Center, Beijing 100094, China. [3] Preclinical Medical School, Beijing University of Chinese Medicine, Beijing 100019, China. [4] State Key Discipline of Infectious Diseases, Shenzhen Third People's Hospital, Shenzhen 518116, China. [5] Shanghai Engineering Research Center of Tooth Restoration and Regeneration, School and Hospital of Stomatology, Tongji University, Shanghai 200072, China. * These authors contributed equally to this work. Correspondence and requests for materials should be addressed to Y.L. (email: yingxianli@aliyun.com) or to Y.S. (email: sunyao919@126.com) or to X.W. (email: txg_wang@jnu.edu.cn).

Myogenesis in the adult is a highly regulated process that begins with the activation and differentiation of muscle stem cells (MuSCs), and then proceeds with cell proliferation, migration and fusion. With the further accretion of nuclei, terminal differentiation is initiated to form multi-nucleated myotubes with the capacity to contract[1,2]. Long noncoding RNAs (lncRNAs), commonly defined as transcribed RNAs of more than 200 nucleotides with no coding potential, are involved in numerous important biological processes[3,4]. To date, only a limited number of lncRNAs have been well characterized, with a diverse array of mechanisms identified, including roles as signalling molecules[5–7], scaffolds[8], guides[9] and decoys[10,11]. It is worth noting that some lncRNAs have been determined to regulate myogenesis[12,13]. For example, noncoding RNA steroid receptor RNA activator (SRA) was reported to promote myogenic differentiation by regulating the transcriptional activity of MyoD[14,15]. LncRNA H19 has a critical role in skeletal muscle differentiation and regeneration, which is mediated by miR-675-3p and miR-675-5p, which are encoded within H19 (ref. 16). MUNC, located upstream of MyoD and specifically expressed in skeletal muscle, is a lncRNA that can promote myogenesis by regulating MyoD expression[17]. Similarly, LncMyoD, activated by MyoD, plays an important role in promoting myogenesis and skeletal muscle regeneration[18]. lnc-MD1 (ref. 19), Glt2/Meg3 (ref. 13), lnc-YY1 (ref. 20) and lncRNA-Dum[21] are also believed as important positively regulators of myogenesis. In contrast, recent studies have shown that certain lncRNAs negatively regulate myogenesis. For instance, m½-sbsRNA inhibits myogenesis via reducing TRAF6 by Staufen-mediated messenger RNA decay[22]. Yam (YY1-associated muscle lncRNAs)[23], H19 (ref. 24), lnc-31 (ref. 25) and Sirt1 AS lncRNAs[22,26] were reported to inhibit myogenic differentiation.

Recently, a class of lncRNAs, referred to as competing endogenous RNAs (ceRNAs), has been characterized[19,24,27–29]. ceRNAs protect mRNAs by acting as molecular sponges for microRNAs (miRNAs) that specifically repress the target mRNAs[30–32]. For instance, lnc-MD1, the first identified ceRNA involved in myogenesis, has been shown to control muscle cell differentiation by competing for the binding of miR-133 and miR-135 (ref. 19). Metastasis-associated lung adenocarcinoma transcript 1 also contains a functional miR-133 target site and can modulate myoblast differentiation by competing for miR-133 (ref. 33). In addition, H19 has been demonstrated to act as a molecular sponge regulating let-7 to control skeletal muscle differentiation[24]. Although functions of these lncRNAs have been partially identified *in vitro* and *in vivo*, most of their roles for myogenesis are still waiting for disclosing. Thus, the aim of our investigation is to explore the role of lncRNAs in regulating myogenic differentiation of adult skeletal MuSCs and skeletal muscle development.

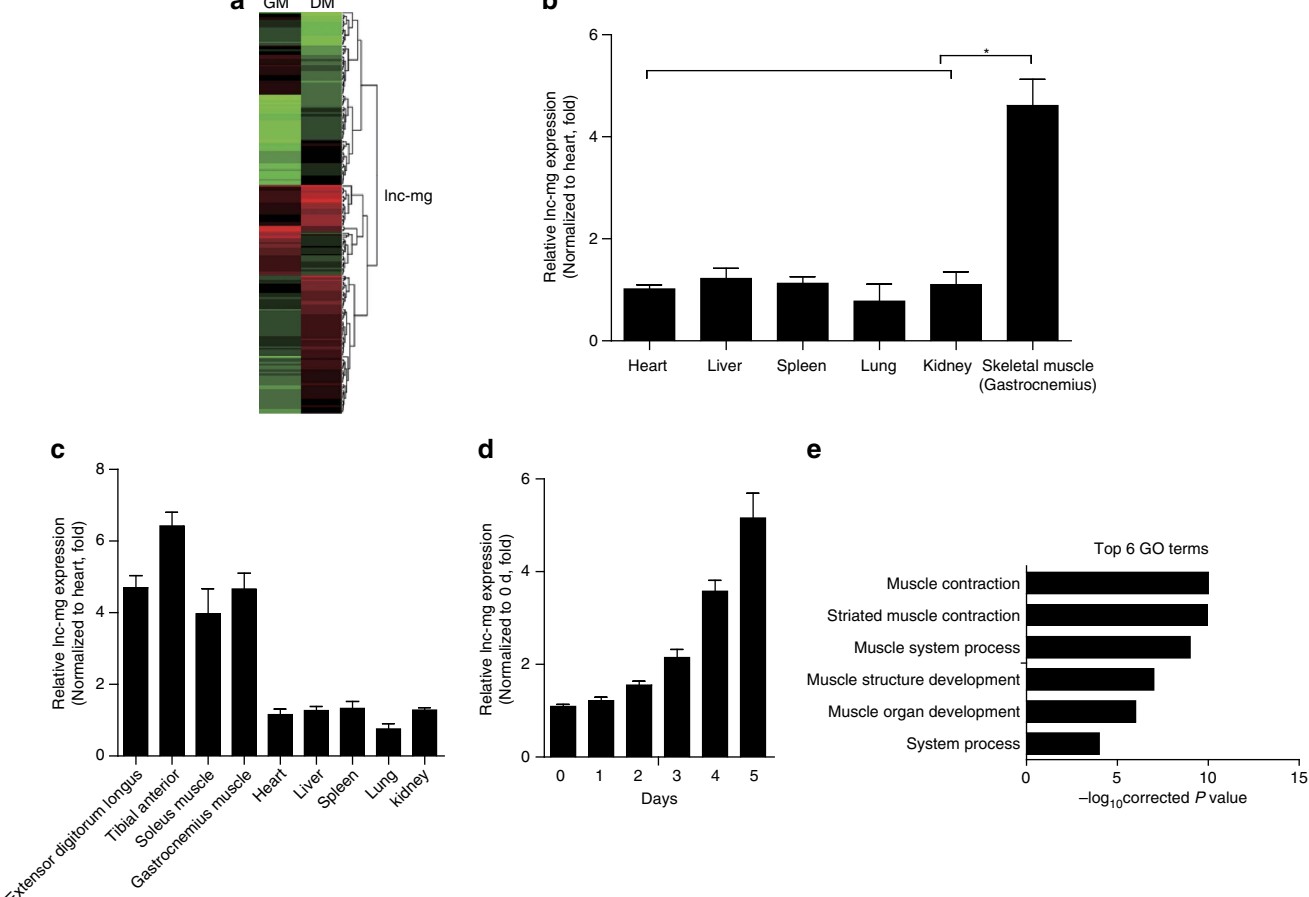

**Figure 1 | Skeletal muscle-enriched lnc-mg is induced during myogenesis. (a)** Microarray heat map of annotated lncRNAs in undifferentiated MuSCs (growth media, GM) and 5-day differentiated MuSCs (differentiation media, DM). **(b)** Real-time PCR analysis of lnc-mg expression in mouse tissues. Mean values ± s.e.m., $n = 6$, *$P < 0.05$. Mice were 8 weeks old, three for male, three for female. **(c)** Real-time PCR analysis of lnc-mg expression in mouse skeletal muscles and other major organs. Mean values ± s.e.m., $n = 6$. Mice were 8 weeks old, three for male, three for female. **(d)** Real-time PCR analysis of lnc-mg expression in MuSCs during 5 days of differentiation. Mean values ± s.e.m., $n = 3$. **(e)** lnc-mg associated gene subsets were determined by Gene Ontology analysis. The data statistical significance is assessed by Student's *t*-test.

In this study, we describe the function of a myogenesis-associated lncRNA (lnc-mg) in mice. lnc-mg is induced during myogenesis and is required during MuSC differentiation. Our study further reveals that lnc-mg promotes myogenesis and enhances muscle mass *in vivo*. In addition, we demonstrate that lnc-mg results in decreased miR-125b by acting as a molecular sponge *in vitro* and *in vivo*, which subsequently controls the expression of insulin-like growth factor 2 (*Igf2*)[34], a critical regulator of skeletal myogenesis.

## Results

**lnc-mg is induced in myogenesis.** To identify functional lncRNAs correlating with myogenesis, we isolated and induced mouse skeletal MuSCs to differentiation (Supplementary Fig. 1a). Microarray data from the original and differentiated MuSCs revealed that 70 lncRNAs were upregulated and 12 were downregulated during this change (Fig. 1a and Supplementary Table 1). Among the increased lncRNAs, we identified a lncRNA (named lnc-mg) enriched in skeletal muscle (Fig. 1b and Supplementary Fig. 1b). To validate whether lnc-mg expressed differently in various types of muscles, we examined the levels of lnc-mg in different types of muscles. It is found that the expression levels of lnc-mg have only a little difference among different types of muscles, while higher than in other tissues (Fig. 1c and Supplementary Fig. 1c). The general information and sequence of lnc-mg are supplied in Supplementary Fig. 1d,e.

In addition, lnc-mg has a polyA tail and a 5′-cap structure (Supplementary Fig. 1f) but without coding capacity (Supplementary Fig. 1g). Consistent with the microarray data, lnc-mg is shown to be induced in MuSCs differentiation (Fig. 1d). Interestingly, in Gene Ontology analysis, lnc-mg-related genes are mainly clustered into muscle contraction and muscle system process classification categories (Fig. 1e).

**lnc-mg promotes MuSCs differentiation.** To investigate the role of lnc-mg during myogenesis *in vitro*, we used RNA interference to knock down lnc-mg and an expression vector to overexpress lnc-mg in MuSCs. Successful knockdown of lnc-mg (Fig. 2a) results in significant inhibition of MuSCs differentiation proved by the reduced expression of myosin heavy chain (MyHC) (Fig. 2b), decreased number of positive myotubes (Fig. 2c) and downregulated expression of myogenic marker genes *Myod* and *Myog* (Fig. 2d). Moreover, overexpression of lnc-mg (Fig. 2e) accelerates the differentiation of MuSCs with increased MyHC immunostaining (Fig. 2f), increased myotubes numbers (Fig. 2g) and upregulated *Myod* and *Myog* expression (Fig. 2h).

**lnc-mg$^{skl-/-}$ mice result in muscle atrophy and weakness.** We next sought to determine the function of lnc-mg during myogenesis *in vivo*. Consequently, lnc-mg skeletal muscle-conditional knockout mice were generated (Supplementary Fig. 2a,b). Compared with control *floxp* mice (lnc-mg$^{fl/fl}$), morphometric

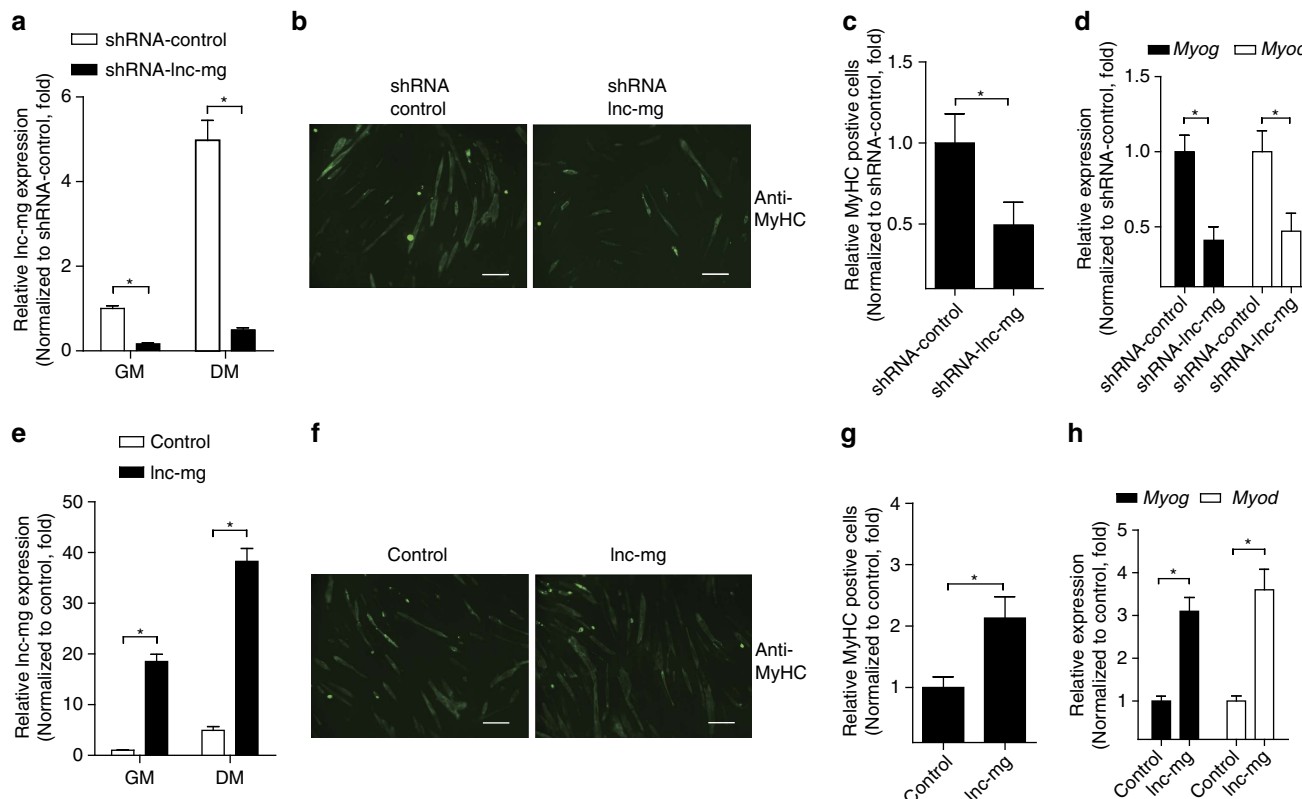

**Figure 2 | lnc-mg promotes MuSC differentiation *in vitro*.** (**a**) Real-time PCR analysis of lnc-mg expression in MuSCs transfected with control shRNA or lnc-mg shRNA. (**b**) MyHC immunostaining of MuSCs transfected with control shRNA or lnc-mg shRNA then cultured in differentiation medium (DM) for 5 days. Scale bar, 40 μm. (**c**) Comparison of cultured MyHC-positive cells transfected with control shRNA or lnc-mg shRNA. (**d**) Real-time PCR analysis of *Myog* and *Myod* expression in MuSCs transfected with control shRNA or lnc-mg shRNA then cultured in DM for 5 days. (**e**) Real-time PCR analysis of lnc-mg expression in MuSCs transfected with control vector or lnc-mg vector. (**f**) MyHC immunostaining of MuSCs transfected with control vector or lnc-mg vector then cultured in DM for 5 days. Scale bar, 40 μm. (**g**) Comparison of cultured MyHC-positive cells transfected with control vector or lnc-mg vector. (**h**) Real-time PCR analysis of *Myod* and *Myog* expression in MuSCs transfected with control vector or lnc-mg vector then cultured in DM for 5 days. All data are shown as mean values ± s.e.m., n = 4, *P < 0.05. The data statistical significance is assessed by Student's *t*-test. Transient transfection of MuSCs with control shRNA or lnc-mg shRNA, control vector or lnc-mg vector by using Lipofectamine 3000 reagent.

analysis shows the lower weight of gastrocnemius muscle (GAS), soleus muscle (SOL), extensor digitorumlongus (EDL), tibial anterior (TA) in lnc-mg skeletal muscle-conditional knockout mice ($lnc\text{-}mg^{skl-/-}$) (Fig. 3a). Moreover, the cross-sectional area of muscle fibres in the GAS is smaller in $lnc\text{-}mg^{skl-/-}$ mice (Fig. 3b,c). Measurement of the mean diameter of muscle fibres reveals that $lnc\text{-}mg^{skl-/-}$ mice have a larger number of thinner fibres (Fig. 3d). In addition, the force and specific tetanic force (Fig. 3e), as well as muscle performance in forced treadmill running tests (Fig. 3f) are all reduced in $lnc\text{-}mg^{skl-/-}$ mice.

**lnc-mg $TG$ mice result in muscle hypertrophy.** To further confirm the role of lnc-mg in myogenesis *in vivo*, lnc-mg skeletal muscle-specific transgenic mice ($TG$) were established. The GAS, SOL, EDL and TA weight are higher in $TG$ mice than their wild-type siblings ($WT$) (Fig. 4a). The $TG$ mice also show visibly larger cross-sectional area of muscle fibres (Fig. 4b,c). Consistently, further measurement reveals that force and the specific tetanic force are higher (Fig. 4d) and muscle performance is improved

in $TG$ mice (Fig. 4e). Furthermore, to extend these findings in muscle formation, we explored the function of lnc-mg in denervated skeletal muscular atrophy mice. It shows that the reduction of muscle mass in the $TG$ is weaker than that in $WT$ mice upon denervation, as proved by the larger cross sectional area of dystrophin-positive muscle fibres (Fig. 4f,g).

**lnc-mg acts as a molecular sponge for miR-125b *in vitro*.** lnc-mg locates in both cytoplasm and in nucleus (Supplementary Fig. 3a,b), and the amount of lnc-mg in cytoplasm increases significantly in differentiated myoblast C2C12 cells (Supplementary Fig. 3c). We speculated that lnc-mg may function as a ceRNA, leading to the liberation of corresponding miRNA-targeted transcripts. To test this hypothesis, we used microarray analysis to detect miRNAs expression in C2C12 cells with lnc-mg overexpression or lnc-mg knockdown (Fig. 5a and Supplementary Table 3). Among the candidate miRNAs, we determined that miR-125b was substantially down regulated when lnc-mg was overexpressed, while substantially upregulated when lnc-mg

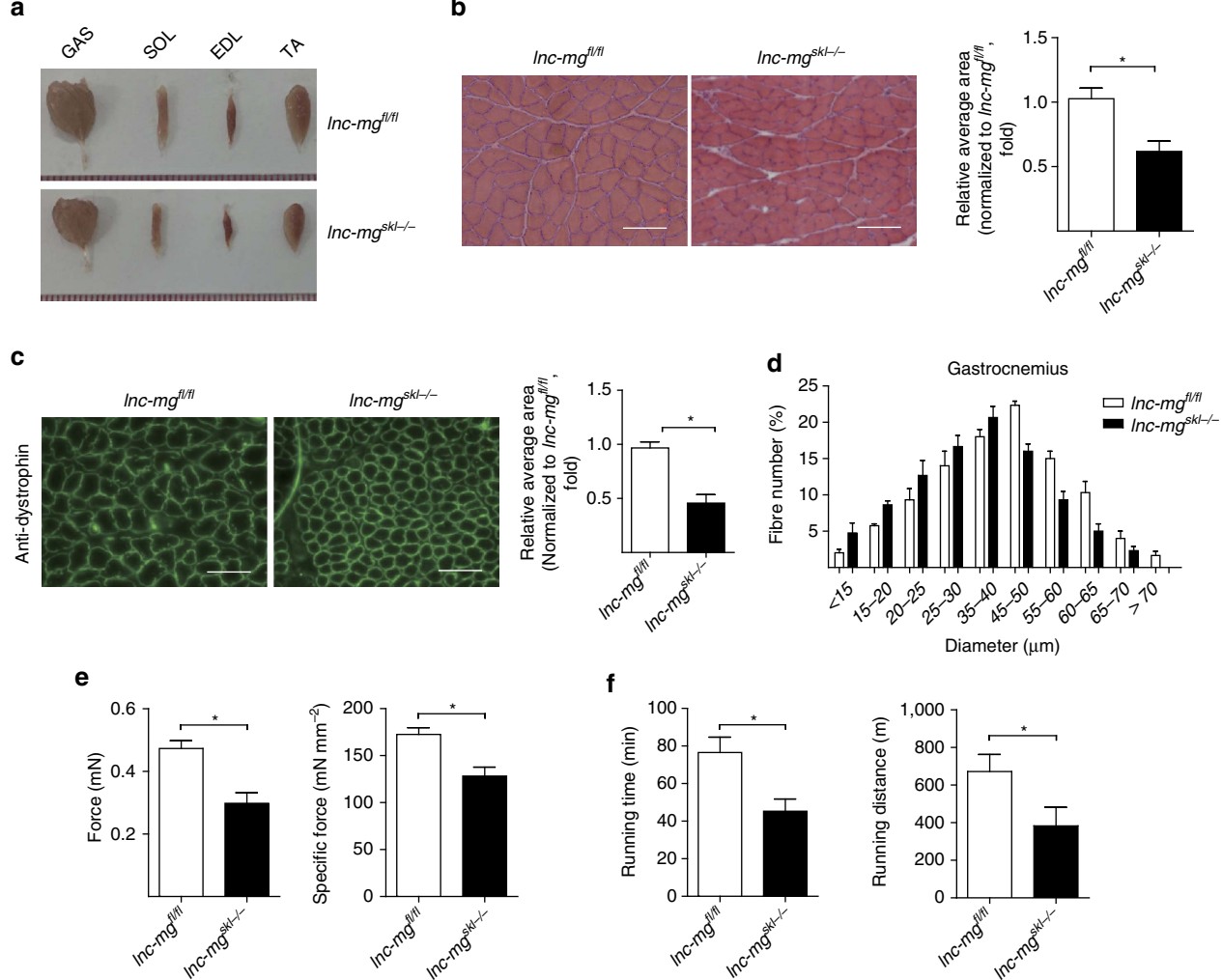

**Figure 3 | Morphological and functional changes in $lnc\text{-}mg^{skl-/-}$ mice.** (**a**) Representative graphs of GAS, SOL, EDL and TA from control Flox mice ($lnc\text{-}mg^{fl/fl}$) and lnc-mg skeletal muscle-conditional knockout mice ($lnc\text{-}mg^{skl-/-}$). (**b**) Histological analysis of GAS sections stained with haematoxylin and eosin. The average muscle fibre cross-sectional areais shown on the right. Mean values ± s.e.m., $n = 6$, *$P < 0.05$. Scale bar, 50 μm. (**c**) Immunostaining of mouse GAS muscle fibre cross-sectional area by anti-dystrophin antibody. Relative average area is shown on the right. Mean values ± s.e.m., $n = 6$, *$P < 0.05$. Scale bar, 50 μm. (**d**) Size distribution and mean diameter of muscle fibres were measured in gastrocnemius from $lnc\text{-}mg^{fl/fl}$ mice and $lnc\text{-}mg^{skl-/-}$ mice. Mean values ± s.e.m., $n = 6$. (**e**) Force measurements performed on GAS from $lnc\text{-}mg^{fl/fl}$ mice and $lnc\text{-}mg^{skl-/-}$ mice during tetanic contraction. Mean values ± s.e.m., $n = 6$, *$P < 0.05$. (**f**) Muscle performance was measured using forced treadmill running to exhaustion. Mean values ± s.e.m., $n = 6$, *$P < 0.05$. The data statistical significance is assessed by Student's *t*-test. Mice were 8 weeks old, three for male, three for female.

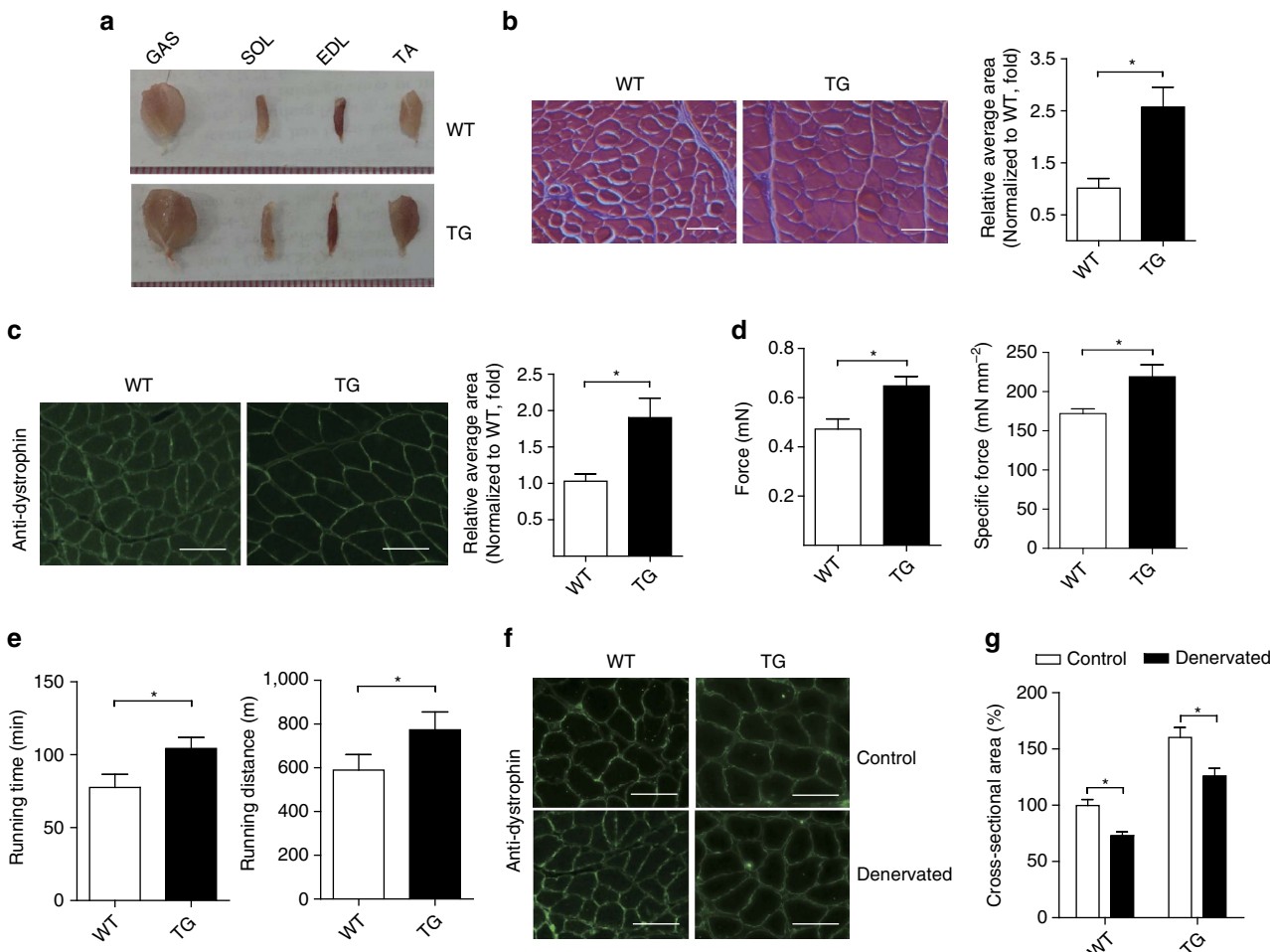

**Figure 4 | lnc-mg *TG* mice induce muscle hypertrophy.** (**a**) Representative graphs of GAS, SOL, EDL and TA from WT mice (*WT*) and lnc-mg skeletal muscle-specific TG mice (*TG*). (**b**) Histological analysis of GAS sections from *WT* and *TG* mice stained with haematoxylin and eosin. Relative average area is shown on the right. Mean values ± s.e.m., n = 6, *P < 0.05. Scale bar, 50 μm. (**c**) Cross-sectional area of GAS muscle fibre was stained by anti-dystrophin antibody. Relative average area is shown on the right. Mean values ± s.e.m., n = 6, *P < 0.05. Scale bar, 50 μm. (**d**) Force measurements performed on GAS from *WT* and *TG* mice during tetanic contraction. Mean values ± s.e.m., n = 6, *P < 0.05. (**e**) Muscle performance was measured using forced treadmill running to exhaustion. Mean values ± s.e.m., n = 6, *P < 0.05. (**f**) Dystrophin staining of GAS muscle fibres from *WT* and *TG* mice denervated and examined 14 days later. Scale bar, 50 μm. (**g**) Cross-sectional area of GAS from *WT* and *TG* mice after denervation. Mean values ± s.e.m., n = 6, *P < 0.05. The data statistical significance is assessed by Student's *t*-test. Mice were 8 weeks old, three for male, three for female.

was knocked down (Fig. 5b) with a dose-dependent manner (Supplementary Fig. 4a,b). Moreover, bioinformatics analysis reveals a predicted miR-125b response element resides in the lnc-mg transcript. To validate that lnc-mg was indeed targeted by miR-125b, WT and mutant miR-125b were synthesized (Fig. 5c), and luciferase reporters containing a WT or mutant target site from lnc-mg were also constructed (Fig. 5d). Only WT miR-125b (AgomiR-125b) significantly reduces luciferase activity for the WT lnc-mg reporter (Fig. 5e) and only WT lnc-mg targeting site is recognized by AgomiR-125b (Fig. 5f). Further studies show that the luciferase activity of WT lnc-mg reporter is specifically increased upon reduction of endogenous miR-125b levels with Antagomir-125b (Fig. 5g,h). All these data demonstrate that lnc-mg contains functional miR-125b binding sites. For further confirmation, Ago2 immunoprecipitation and biotin-labelled miR-125b capture followed by real-time PCR confirm the interaction between miR-125b and lnc-mg (Fig. 5i,j).

Previous research has shown that miR-125b could negatively modulate myoblast differentiation by directly targeting *Igf2* (ref. 34). Thus, we examined *Igf2* expression in C2C12 cells with lnc-mg overexpression or knockdown. It reveals that *Igf2* is

downregulated by lnc-mg knockdown and upregulated by lnc-mg overexpression, as assessed by western blotting (Fig. 5k), and the ability of lnc-mg to modulate *Igf2* expression in cell supernatant is further verified by enzyme-linked immunosorbent assay (ELISA) (Fig. 5l). Furthermore, overexpression of lnc-mg leads to the increased enrichment of Ago2 on lnc-mg, while substantially decreased enrichment on *Igf2* (Supplementary Fig. 4c). For further confirmation, luciferase reporter assay shows that the luciferase activity of *Igf2* 3′-untranslated region reporters is increased upon WT lnc-mg overexpression but not upon miR-125b binding site mutated lnc-mg (Supplementary Fig. 4d). Streptavidin capture analysis further suggests that the binding enrichment of miR-125b on *Igf2* decreases with overexpression of WT lnc-mg, but not with miR-125b-binding site mutated lnc-mg (Supplementary Fig. 4e).

**lnc-mg modulates miR-125b by functioning as a ceRNA *in vivo*.** To determine whether lnc-mg functions as a molecular sponge for miR-125b *in vivo*, we performed real-time PCR analysis of miR-125b levels in mice. Compared with *lnc-mg*^fl/fl^ mice, the relative expression of miR-125b is higher in GAS from

$lnc\text{-}mg^{skl-/-}$ mice (Fig. 6a). Western blotting and ELISA assay indicate that, conversely, the levels of Igf2 protein are lower in GAS and serum from $lnc\text{-}mg^{skl-/-}$ mice compared to $lnc\text{-}mg^{fl/fl}$ mice (Fig. 6b,c). In addition, miR-125b levels are lower and the Igf2 protein levels are higher in GAS and serum from lnc-mg *TG* mice compared to *WT* mice (Fig. 6d–f).

## Discussion

In recent years, several groups have described lncRNAs containing miRNA binding sites that function as molecular sponges to effectively inhibit miRNA function[35,36]. lnc-RoR controls self-renewal of human embryonic stem cells by protecting OCT4, SOX2 and NANOG transcripts from miR-145-mediated

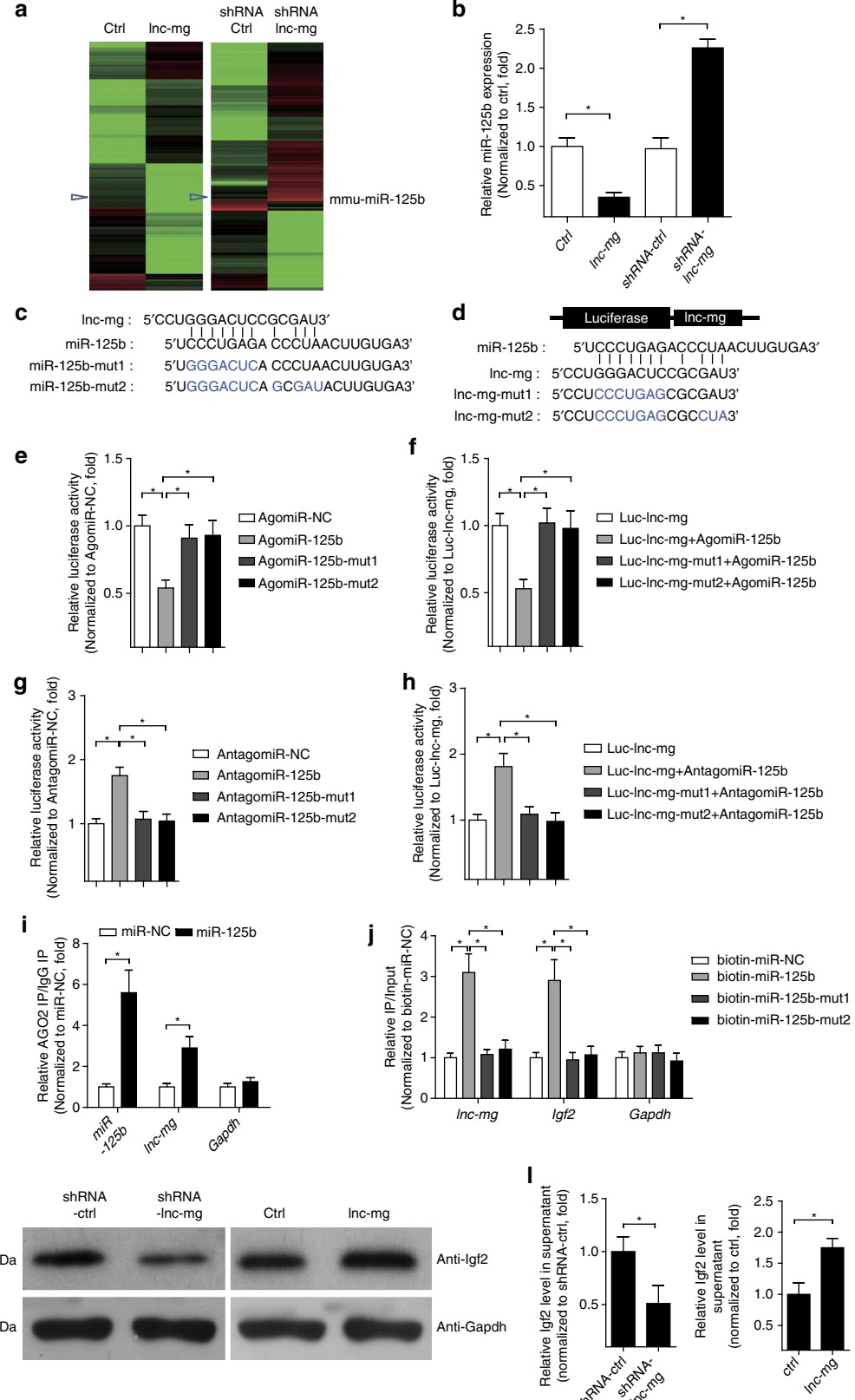

suppression[37]. Another lncRNA, PTENP1, regulates the expression of the tumor suppressor phosphatase and tensin homologue by competing for shared miRNAs[38]. In addition, a novel class of RNA molecules, termed circular RNAs, has also been demonstrated to function as efficient miRNA sponges[39,40].

Although several lncRNAs have been shown to have roles in skeletal muscle cell differentiation and muscle development *in vitro*[16,41,42], little is known about their function during myogenesis *in vivo*, with the exception of a putative lncRNA that actually encodes for a micropeptide[43,44]. In this study, we identify a skeletal muscle-enriched lncRNA (named lnc-mg), that promotes myogenesis *in vitro* and *in vivo*. This study provides comprehensive functional and mechanistic characterization of lnc-mg, using both lnc-mg skeletal muscle-conditional knockout mice and skeletal muscle-specific TG mice. By functioning as a ceRNA, lnc-mg blocks miR-125b to control Igf2 protein level *in vitro* and *in vivo*. Although we observe that transgene induces hypertrophy, the rescue effect of lnc-mg on muscle loss needs to be carefully investigated, for we find that muscle loss is not significant changed after denervation in *TG* mice.

In this study, we demonstrate that miR-125b expression is notably suppressed when lnc-mg is overexpressed. Interestingly, miR-125b has been reported to negatively modulate myoblast differentiation and its expression is known to be down-regulated during myogenesis[34]. In addition, Igf2 (refs 34,45,46), a key regulator of myogenesis, has been confirmed to serve as a target of miR-125b[47]. We demonstrate that miR-125b levels in tissue and cells are down-regulated when lnc-mg is overexpressed, resulting in increased Igf2 protein to enhance myogenesis. The graphic abstract of lnc-mg regulating myogenesis is shown in Fig. 7. In conclusion, lnc-mg is a key myogenesis enhancer by functioning as a ceRNA for miR-125b controlling protein abundance of Igf2.

## Methods

**Mice.** Animal protocol are approved by the Animal Ethics Committee of Peking Union Medical College, Beijing, China. Eight-week old, C57B/6J mice were used in our study, three for male and three for female in each group.

***In vitro* cell culture and differentiation.** Mouse skeletal MuSCs were isolated from 10-day-old C57B/6J mice according to the previously described procedure[48]. Briefly, total hind limb muscles were incubated with muscle dissociation buffer (700–800 U ml$^{-1}$ collagenase II solution prepared in Ham's F-10 wash medium supplemented with 10% horse serum and 1 × penicillin–streptomycin) in 37 °C with agitation and washed with cold wash medium. Next, the cells were centrifuged at 500 $g$ for 5 min at 4 °C. After stocking solution was added to the cells to block collagenase II dispase, cells were incubated in 37 °C with agitation for 30 min. Next, the cell suspensions were filtered through 40 μM cell strainer, then pre-plated for 1 h. Non-adherent cells were centrifuged and separated by antibody staining and cell sorting. The cells were cultured in DMEM medium with 10% fetal bovine serum (Life Technologies), 2 mM L-glutamine, 100 U ml$^{-1}$ penicillin and 100 U ml$^{-1}$ streptomycin at 37 °C in a 5% $CO_2$ incubator. To induce differentiation, MuSCs were seeded in 24-well plates and after cells reached 80% confluence the medium was changed by DMEM containing 5% horse serum (Life Technologies). C2C12 myoblasts were obtained from ATCC, cultured in DMEM with 10% fetal bovine serum (growth media) and induced myogenic differentiation by switching the medium to DMEM containing 2% horse serum (differentiation media).

**Real-time PCR analysis.** Total RNA from tissues or cells was extracted in TRIzol Reagent (Life Technologies) according to the manufacturer's instructions. RNA (1 μg) was reverse-transcribed by using the PrimeScript RT reagent Kit with gDNA Eraser (Perfect Real Time) (Takara). One microlitre of a 1:5 dilution of the synthesized complementary DNA was used for real-time PCR analysis. The relative abundance of the mRNAs was determined using SYBR Premix Ex Taq II (TliRNaseH Plus) (Takara) according to the manufacturer's instructions. The following thermal settings were used: 95 °C for 30 s followed by 40 cycles of 95 °C for 5 s and 60 °C for 30 s. Primers[49] used for real-time PCR were listed in Supplementary Table 2. Relative expression values were calculated using the comparative threshold cycle ($\Delta\Delta$CT) method in accordance with the MIQE guidelines[50,51].

**Cell transfection.** Transient transfection of cells with miRNA mimic, short hairpin RNA (shRNA) or DNA plasmids was performed in 24-well plates using Lipofectamine 3000 reagent[52] (Life Technologies). For lnc-mg functional analyses, lnc-mg expression plasmid (500 ng per well) or empty plasmid (500 ng per well) and shRNA control (100 nM) or shRNA lnc-mg (100 nM) were transfected into cells in culture medium and then harvested for further detection. For luciferase experiments, miRNA agomir (100 nM) or miRNA antagomir (100 nM) and psiCHECK-2 (500 ng per well) containing the WT or mutated sequence of lnc-mg were transfected into cells. Cells were harvested for the dual-luciferase assay 24 h after transfection.

**Immunofluorescence staining.** Cells were fixed in 4% paraformaldehyde for 15 min and permeabilized in 0.25% Triton X-100 for 10 min at room temperature. The cells were blocked in 1% BSA for 30 min at room temperature and then incubated with primary antibody to MyHC (MF20, 1:400, Developmental Studies Hybridoma Bank, University of Iowa) at 4 °C overnight with gentle shaking, followed by incubation with fluorescein isothiocyanate-conjugated secondary antibody (Cell Signaling Technology, 1:100) at room temperature for 1 h, with thrice PBS washes after each antibody incubation. Nuclei were counter-labeled with DAPI. The immunofluorescence images were visualized with a fluorescence microscope (Leica image analysis system, Model Q500MC).

**Skeletal muscle-conditional lnc-mg knockout mice model.** For the generation of lnc-mg$^{flox/flox}$ mice, targeting vector was constructed by inserting a Frt-flanked neomycin cassette upstream and two loxP sites downstream of the first exon of lnc-mg and then electroporating into embryonic stem cells from C57BL/6J mice (conducted by Beijing Biocytogen Co. Ltd). Skeletal muscle-conditional lnc-mg knockout mice were generated by crossbreeding of lnc-mg$^{flox/flox}$ mice with MCK-Cre mice (from Jackson Laboratory).

**Skeletal muscle-specific lnc-mg TG mice model.** A plasmid containing the MEF2-myogenin promoter (kind gift from Prof. Eric N. Olson, University of Texas Southwestern Medical Center) to drive lnc-mg-specific expression in skeletal muscle was used to generate skeletal muscle-specific lnc-mg TG mice. A fragment of the MEF2-myogenin-promoter-lnc-mg was purified and microinjected into

**Figure 5 | lnc-mg functions as a ceRNA for miR-125b *in vitro*.** (**a**) Microarray heat map of differential expressed miRNAs in C2C12 myoblasts transfected with control vector (ctrl), lnc-mg vector (lnc-mg), control shRNA (shRNA ctrl) and lnc-mg shRNA (shRNA lnc-mg), respectively. (**b**) Real-time PCR analysis of miR-125b expression in C2C12 myoblasts transfected with control vector (ctrl), lnc-mg vector, control shRNA and lnc-mg shRNA, respectively. (**c**) The lnc-mg site is predicted to be a target of miR-125b. Two seed sequence mutants of miR-125b (miR-125b-mut1 and miR-125b-mut2) are shown below. The blue letters represent the mutant sites. (**d**) Luciferase reporter constructs: WT lnc-mg and two lnc-mg (lnc-mg-mut1 and lnc-mg-mut2) with mutations in the miR-125b-binding sites were inserted into psiCHECK-2 vector. The blue letters represent the mutant sites. (**e**) The relative luciferase activity of psiCHECK-2 containing WT lnc-mg co-transfected with AgomiR-NC, AgomiR-125b or mutated AgomiR-125bin C2C12 cells. (**f**) The relative luciferase activity of psiCHECK-2 containing WT or mutated lnc-mg co-transfected with AgomiR-125b in C2C12 cells. (**g**) The relative luciferase activity of psiCHECK-2 containing WT lnc-mg co-transfected with AntagomiR-NC, Antagomir-125b or mutated Antagomir-125b. (**h**) The relative luciferase activity of psiCHECK-2 containing WT or mutated lnc-mg co-transfected with Antagomir-125b. (**i**) Ago2 immunoprecipitation was performed in C2C12 myoblasts transfected with control miRNA (miR-NC) or miR-125b, followed by real-time PCR, to detect lnc-mg associated with Ago2. (**j**) Streptavidin capture was performed for C2C12 myoblasts transfected with biotin-miR-NC, biotin-miR-125b or mutated biotin-miR-125b, followed by real-time PCR to detect lnc-mg, Igf2 mRNA and Gapdh mRNA levels. (**k**) Western blot analysis of Igf2 protein levels in C2C12 myoblasts transfected with control shRNA or lnc-mg shRNA, control vector (ctrl) or lnc-mg vector. (**l**) ELISA analysis of secreted Igf2 protein levels in supernatant from C2C12 myoblasts transfected with control shRNA or lnc-mg shRNA, control vector (ctrl) or lnc-mg vector. All data are shown as mean values ± s.e.m., n = 3, *P < 0.05. The data statistical significance is assessed by Student's t-test. Transient transfection of C2C12 cells using Lipofectamine 3000 reagent.

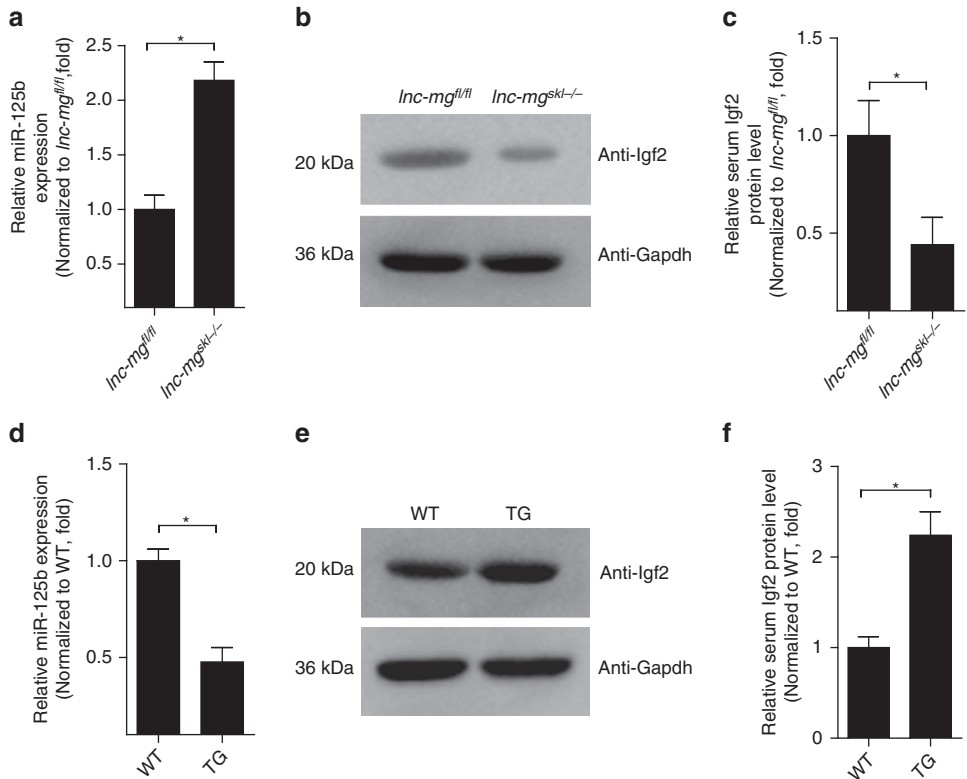

**Figure 6 | lnc-mg functions as a ceRNA for miR-125b in vivo. (a)** Real-time PCR analysis of miR-125b expression in GAS from *lnc-mg^fl/fl^* mice and *lnc-mg^skl−/−^* mice. Mean values ± s.e.m., $n = 6$, *$P < 0.05$. **(b)** Western blot analysis of Igf2 protein levels in GAS from *lnc-mg^fl/fl^* mice and *lnc-mg^skl−/−^* mice. **(c)** ELISA analysis of Igf2 protein levels in serum from *lnc-mg^fl/fl^* mice and *lnc-mg^skl−/−^* mice. Mean values ± s.e.m., $n = 6$, *$P < 0.05$. **(d)** Real-time PCR analysis of miR-125b expression in GAS from *WT* and *TG* mice. Mean values ± s.e.m., $n = 6$, *$P < 0.05$. **(e)** Western blot analysis of Igf2 protein levels in GAS from *WT* and *TG* mice. **(f)** ELISA analysis of Igf2 protein levels in serum from *WT* and *TG* mice. Mean values ± s.e.m., $n = 6$, *$P < 0.05$. The data statistical significance is assessed by Student's *t*-test. Mice were 8 weeks old, three for male, three for female.

C57BL/6J mouse oocytes, and the oocytes were then surgically transferred into pseudopregnant C57BL/6J dams by Cyagen Biosciences Inc (Guangzhou, China).

**Denervated skeletal muscular atrophy mice.** To denervate the GAS muscle, mice were deeply anesthetized by intraperitoneal injection of 2 ml kg$^{-1}$ chloral hydrate, and then a dorsolateral skin incision was made on the lower hind limb to excise about 1.5 cm of the sciatic nerve. The nerve was sewn on the muscle membrane to prevent it from reconnecting. Control group was performed by exposing the sciatic nerve but keeping it intact and then stitching the incision.

**Muscle fibrosis staining.** Fibrosis staining was performed according to the published method[53] with slight modification. Briefly, paraffin sections of GAS were incubated with 5% BSA for 30 min, with primary antibodies against dystrophin (Abcam, 1:100) at 4 °C overnight and then with fluorescein isothiocyanate-conjugated secondary antibody (Cell Signaling Technology, 1:100) for 1 h at room temperature. Images were visualized and captured with a fluorescence microscope (Leica image analysis system, Model Q500MC).

**Cross-section area and diameter of muscle fibres.** The measurement of cross-section area was performed according to the published method with slight modification[54]. The cross-section area of each muscle (fibres number, $n = 500$) in four fields from each animal of six mice (8-week old C57BL/6J mice, three for male, three for female) were randomly chosen and determined using the ImageJ programme, and then calculated the mean cross-section area of each group. Fibre diameter was calculated as the caliper width perpendicular to the longest chord of each fibre. The total fibre number was calculated using an image of $\times 20$ magnification from the entire field of muscle section, which was randomly chosen.

**Hematoxylin and eosin staining.** Histological analysis of muscle sections was performed essentially according to the published method[55]. The GAS was fixed in 4% paraformaldehyde, processed and embedded in paraffin prior to sectioning (10 μm) and staining. The tissues were fixed in 1% osmium tetraoxide in 0.1% M-cacodylate buffer for 1 h at 4 °C, then dehydrated and embedded in a pure epoxy

resine, which became solid after 48 h at 60 °C. Semi-thin sections were made, and the epoxy resine eluted and stained with haematoxylin and eosin.

**Measurement of muscle weight and muscle force.** The SOL, EDL, GAS and TA of 8-week-old (three for male and three for female) *lnc-mg^Skl−/−^*, *WT* and *TG* mice were harvested and weighed by electronic 1/10,000 scale. The muscle force measurement was performed essentially according to the published method[56]. Mice were anaesthetized via intraperitoneal injection of a cocktail containing 25 mg ml$^{-1}$ ketamine, 2.5 mg ml$^{-1}$ xylazine and 0.5 mg ml$^{-1}$ acepromazine at 2.5 ml per gram body weight. The entire GAS was isolated and preserved in Ringer's solution which was continuously aerated with 95% $O_2$ and 5% $CO_2$, and maintained at 37 °C. The distal end of GAS was connected to an isometric transducer. The GAS was stimulated with electrical stimulation of 100 Hz. Optimal muscle length was multiplied by 0.85, to calculate the optimal fibre length. Tetanic contractions induced by optical and electrical stimulation were 2 s long with an interval of 3 min between stimulations. Maximal force (M) was analysed for single twitch contractions. Specific force was normalized for muscle cross-sectional area (CSA), which was calculated by mass (mg)/fibre length (mm)*1.06 (mg mm$^{-3}$). The specific force was calculated by M/CSA.

**Exercise performance test.** An exercise performance test was performed essentially according to the published method[57]. Briefly, mice were first accustomed to treadmill running on a 20° incline and 25 cm s$^{-1}$ belt speed for 3 days. In the first day, 10 min running in the morning and 10 min running in the afternoon was employed at 20 cm s$^{-1}$ belt speed without incline. In the second day, 10° incline 20 cm s$^{-1}$ belt speed in the morning and 10° incline 25 cm s$^{-1}$ in the afternoon was used for each 15 min running. In the third day, 25° incline 25 cm s$^{-1}$ belt speed in the morning and 25° incline 30 cm s$^{-1}$ in the afternoon was put into use for each 20 min running; In the fourth day, mice ran on a 25° incline and 30 cm s$^{-1}$ belts peed for 20 min and then the belt speed was increased by 4 cm s$^{-1}$ every 20 min until the mice were exhausted.

**Microarray assay.** The microarray experiments were performed by RiboBio (Guangzhou, China). Briefly, total RNAs was extracted from cells of lnc-mg overexpression or lnc-mg knockdown by TRIzol Reagent (Life Technologies).

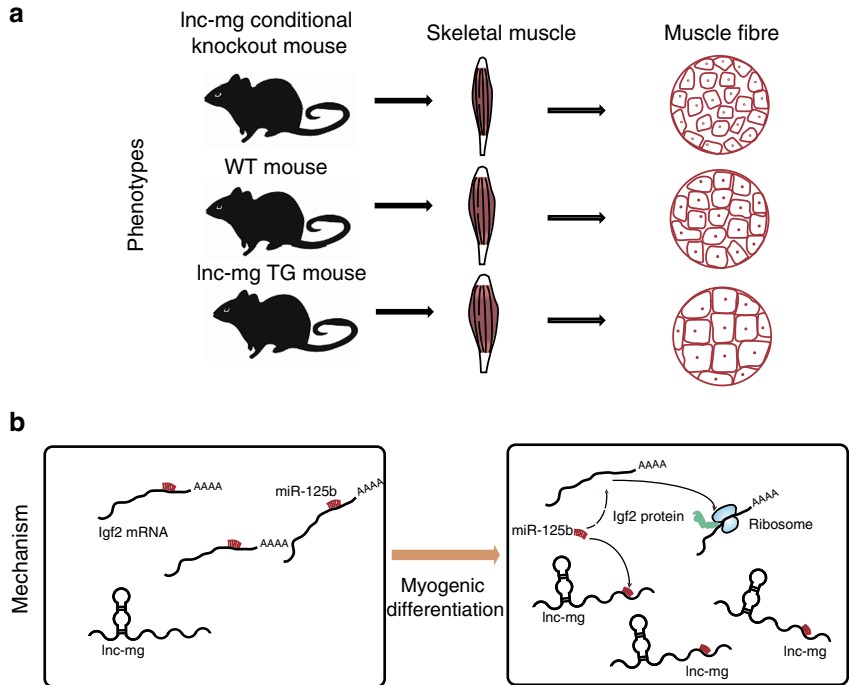

**Figure 7 | Graphical abstract of how lnc-mg regulates myogenesis. (a)** Skeletal muscle specific lnc-mg TG mouse with enhanced myogenesis and skeletal muscle conditional lnc-mg knockout mouse with weakened myogenesis compared to WT mouse. (**b**) Molecular mechanism of lnc-mg: regulating miR-125b to control Igf2 protein level by functioning as a ceRNA.

Then the RNA quality was assessed by formaldehyde agarose gel electrophoresis, quantified spectrophotometrically and Agilent 2200 Bioanalyzer (Agilent, USA). Total RNA (1.5 μg) was labelled on Cy5 using Universal Linkage System (ULS). Then the CustomArray microarray was pre-hybridized in nuclease-free water at 65 °C for 10 min and then loaded the microarray onto the rotisserie in the hybridization oven and incubated at 37 °C for 60 min with gentle rotation. The hybridization solution was prepared with labeled miRNA target and denatured the hybridization solution at 95 °C for 3 min, and then cooled for 20 s on ice. Next, the microarray was loaded with the hybridization solution and incubated at 37 °C for 16 h with gentle rotation. Then, we removed the microarray from the hybridization solution and washed the microarray by using the wash solution to remove non-specific hybridization. Furthermore, we covered the semiconductor microarray surface with the imaging solution and loaded the microarray into the scanner to scan. The data were analysed by Guangzhou RiboBio Co., Ltd.

**Coding capability and the 5′-cap or 3′-poly A detection.** The RNA sequences of *Myh1*, *H19* and lnc-mg were put into the Coding Potential Calculator programme and both *H19* and lnc-mg were predicted to be non-coding RNAs, whereas *Myh1* was identified to code for protein in Method sections. For 3′-poly A detection, total RNAs was extracted from MuSCs in TRIzol Reagent (Life Technologies) according to the manufacturer's instructions. Next, the RNA quality was assessed by formaldehyde agarose gel electrophoresis, quantified spectrophotometrically and Agilent 2200 Bioanalyzer (Agilent). Ribosomal RNA was removed using the Ribo-Zero Magnetic Kits (Illumina) according to the manufacturer's instructions. Then polyA+ RNA fraction and polyA- RNA fraction were isolated by using NEBNext Poly(A) mRNA Magnetic Isolation Module (NEB, NEB E7490S/L). In addition, the amount of lnc-mg was examined in PCR assay with polyA+ RNA fraction and polyA-RNA fraction, respectively. For 5′-cap detection, the experiment was performed using the FistChoice RLM-RACE Kit (Ambion) according to the manufacturer's instructions. In brief, total RNAs from MuSCs was treated with Calf Intestine Alkaline Phosphatase to remove free 5′-P then treated with tobacco acid pyrophosphatase to remove the cap structure and then an RNA adapter oligonucleotide was ligated to the RNA population using T4 RNA ligase. The reverse transcription was performed using primers corresponding to the 5′-rapid amplification of cloned/cDNA ends Adapter sequence provided with the system. PCR amplification was then performed using Taq DNA polymerase (Takara).

**Dual-luciferase assay.** WT lnc-mg or mutated lnc-mg was inserted into psiCHECK-2 (Promega) at the 3′-end of the coding sequence of *Renilla* luciferase then transfected into C2C12 myoblasts. The activity of both luciferases was measured 24 h after transfection. Dual-luciferase assay was performed using the Double-Luciferase Reporter Assay Kit (Promega). Cells were harvested and lysed with Cell Lysis Buffer. Firefly and *Renilla* luciferase activities were evaluated using the Dual-Luciferase Reporter Assay system (Promega). *Renilla* luciferase activity was normalized to the firefly luciferase activity.

**Anti-Ago2 immunoprecipitation.** Cells were harvested 48 h after transfection of FLAG-Ago2 vector. The cells were then lysed by 1 ml of lysis buffer (25 mM Tris-HCl pH 7.4, 150 mM NaCl, 0.5% NP-40, 2 mM EDTA, 1 mM NaF and 0.5 mM dithiothreitol) with RNasin (Takara) and protease inhibitor cocktail (Roche). The supernatant was centrifuged for 30 min at 12,000 g and then 30 μl of anti-FLAG M2 magnetic beads were added (Sigma). After rotating the beads with lysate for 4 h at 4 °C, the beads were washed thrice with washing buffer (50 mM Tris-HCl, 300 mM NaCl pH 7.4, 1 mM MgCl₂, 0.1% NP-40). The RNA was extracted from the remaining beads with TRIzol Reagent (Life Technologies) and evaluated by real-time PCR assay, which is the same to the previous real-time PCR protocol.

**Biotin-labelled miR-125b capture.** Cells were harvested 24 h after transfection and then lysed on ice for 30 min in 250 μl cell lysis buffer (10 mM KCl, 1.5 mM MgCl₂, 10 mM Tris-HCl at pH 7.5, 5 mM dithiothreitol) with RNasin (Takara) and proteinase inhibitor cocktail (Roche). The supernatant was centrifuged for 5 min at 12,000 g, then 500 μl NaCl (1 M) and 30 μl beads (Dynabeads MyOne Streptavidin C1; Life Technologies) were added. Before adding to the supernatant, the beads were washed five times with solution A (0.1 M NaOH, 0.05 M NaCl), then washed thrice with 0.1 M NaCl and were blocked with 1 mg ml⁻¹ BSA (Roche) and 1 mg ml⁻¹ yeast tRNA (Ambion) overnight. The beads were washed five times using washing buffer (5 mM Tris-HCl pH 7.5, 0.5 mM EDTA, 1 M NaCl) after rotating the beads and the lysate for 4 h at 4 °C. RNA was extracted from the remaining beads with TRIzol Reagent (Life Technologies) and evaluated by real-time PCR assay. The entire PCR assay is the same to the previous real-time PCR protocol.

**Western immunoblotting.** Tissues or cells were lysed in RIP buffer on ice for 30 min. The supernatant was centrifuged for 30 min at 12,000 g and 4 °C, ran on SDS–PAGE and then transferred to polyvinylidene difluoride membranes. The membranes were blocked with 5% BSA for 1 h at room temperature and incubated with primary antibody recognizing Igf2 (Abcam, ab9574, 1:1,000) or Gapdh (Abcam, ab8245, 1:2,000) at 4 °C overnight. Incubation with secondary horseradish peroxidase-labelled antibody was carried out for 1 h at room temperature.

**ELISA analysis.** To assess the level of Igf2 in cell supernatant or serum, ELISA assay was performed on microtitre plates coated with antibody that recognizes Igf2 (Abcam, ab9574). The plates were incubated at 4 °C overnight and blocked with 5%

BSA for 2 h at room temperature. The samples were added to the plates and the plates were incubated overnight at 4 °C. Horseradish peroxidase-conjugated secondary antibody (Sigma; 1: 5,000 diluted in 5% BSA) was added and the plates were incubated for 2 h at room temperature. After washing, substrate solution was added, plates were incubated at 37 °C for 30 min and the reactions were stopped with $H_2SO_4$. Optical density values were measured on a micro plate reader (Thermo Fisher) and the assay was calibrated by means of a serially diluted Igf2 protein standard. The excite/emission spetra were detected at 450 nm.

**Data availability.** All relevant data that support the finding of this study are available from the corresponding author upon reasonable request. Microarray-seq data have been deposited in NCBI under following Accession code: GSE93278.

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

## Acknowledgements

This work was supported by National Natural Science Foundation Projects (81370971 to W.X.G., 81470715 to S.Y. and 31325012 to L.Y.X.) Guangdong Natural Science Funds (2014A030313358 and 2015A030313333), Major project in Guangdong Province of science (2014KZDXM011), Science and Technology Planning Project of Guangdong Province (2013B090500105 and 2014A020210015), Guangdong Natural Science Funds for Distinguished Young Scholar (S2013050013880), Key Project of Chinese National Programs for Research and Development(2016YFC1102705), National Key Technology Support Program (2014BAI04B07) and the Recruitment Program of Global Experts S.Y.

## Author contributions

M.Z. and J.L. constructed the $lnc\text{-}mg^{Skl-/-}$ and $TG$ mice, and performed the molecular biological experiments. J.X. and L.Y. isolated and cultured mouse primary skeletal MuSCs, and performed cellular experiments. M.C. and H.S. raised the mice and conducted the animal experiments. X.C., Y.M., S.H., Z.W. and A.H. revised the manuscript. Y.L., Y.S. and X.W. designed this work and wrote this manuscript.

## Additional information

**Competing financial interests:** The authors declare no competing financial interests.

**Publisher's note**: 

