## [Peer review file · Nature Communications]

Reviewers' comments:

Reviewer #1 (Remarks to the Author):

The authors highlight an important, novel, skeletal muscle-enriched lncRNA (named lnc-mg) and its functions and mechanisms during myogenesis. This work has the potential to open up a new area of investigation on lnc-mg, which will also likely be studied in multiple contexts in future studies. Below, I provide a few general comments, as well as comments about specific figures, which I think will improve this manuscript.

General comments:

While the abstract highlights the major, important findings related to lnc-mg in myogenesis, I found the statement that "experiments showing the role of lncRNAs in myogenic differentiation of stem cells and myogenesis in vivo haven't been done" (lines 53-55) to be inaccurate. From my knowledge of the current literature, several lncRNAs have been studied in the context of myogenic differentiation and the majority was shown to promote myogenesis. Moreover several of them are also involved in muscle regeneration (therefore in vivo studies have been done). Similarly, in the introduction, the authors state that ceRNA studies in mice are novel (lines 88-89; 93-94), which is not entirely true given that ceRNAs have been studied in vivo. I recommend the authors revise these statements to less strong statements and instead highlight how their findings on lnc-mg are novel to myogenesis both in vitro and in vivo. I would recommend the authors become familiar with the literature of lncRNAs in myogenesis and cite relevant papers. Showing the readers that the authors are familiar with the latest literature on the topic, as well as showing the major trends known about lncRNAs in myogenesis (therefore broadening the introduction) would be important changes to make in order to create a context in which the novelty of lnc-mg could be better highlighted. Finally, for a more comprehensive analysis of lncRNAs in myogenesis, please see the review article Simionescu-Bankston and Kumar, JMM, 2016 (as well as other excellent published reviews on non-coding RNAs in myogenesis).

Specific comments:

Other than the general comments above, many of the differences shown in the manuscript are not major, which makes it difficult to argue that lnc-mg is a critical regulator of myogenesis. One would expect greater differences in levels of lnc-mg in myogenesis in vitro and in vivo if this were the case. Below, I provide specific examples/comments for particular figures. The authors are advised to take these comments into consideration and fix the issues presented here in all places in the manuscript (even if they are only mentioned in one place in this critique):

In Figure 1b, the increased lnc-mg expression in skeletal muscle is not very large (same comment for figure 2c), posing the question of whether lnc-mg plays a major role here. Nevertheless, the error bars are small and the increase in expression in skeletal muscle appears to be statistically significant visually. The authors need to discuss the statistics utilized to obtain these data, and whether the increase in skeletal muscle is statistically significant. Given that this manuscript focuses on lnc-mg in skeletal muscle, the authors also need to explain why they examined the levels of lnc-mg in so many tissues. I would consider showing a few major tissues here for comparison (for example skeletal muscle, heart, kidney, liver, lung) and have the rest of the data as supplementary.

In Figure 1c, the comments are similar to the ones above. It is currently unclear from the presented data why the expression was analyzed in so many different types of muscles. This is also not explained thoroughly in the text referring to this section of figure 1. Similarly to 1b, the authors could show data in only a few muscles here and the rest be presented as supplementary data. These changes would make the manuscript more focused on the important findings on linc-mg in myogenesis.

The authors also need to explain the meaning behind the data presented in Figure 1c for muscle biology in general. For example, are all the muscles examined here containing the fast-fiber type, and if so, what does it mean that the expression increases in these muscles? Since linc-mg is a novel regulator of myogenesis, a thorough analysis of its expression in both slow and fast muscle types (including TA and soleus) would be beneficial to the muscle field.

With regards to Figure 1d, I believe that generally data showing changes using primary muscle cells is much more relevant and valuable than C2C12 data. In addition, in this case, the data in the two cell types looks exactly the same; therefore I do not believe the C2C12 data add much information. Unless the authors have a very compelling reason for showing these data in both cell types, I recommend removing the C2C12 data.

In Figure 2b, both the phase and fluorescent images of myotubes (both here and everywhere else in the manuscript) are of very poor quality. One cannot discern the myotubes in the phase images, and the number of nuclei in the fluorescent images is very hard to determine. I would have a hard time believing graphs based on these current images. Therefore, very high quality images of muscle cells/myotubes are necessary. In addition to high quality images, a more thorough analysis of myotube size is needed - for example how many myotubes with different numbers of nuclei there are etc. I would also like to see a graph showing indicating that the same cell number was plated/used - showing this information would give confidence that smaller myotubes observed at the end of the experiment are not due to fewer cells being plated in the beginning of the experiment. This analysis is critical to concluding that linc-mg has a role in myogenesis and specifically myotube formation.

Figure 3a shows a very nice decrease in muscle size in the conditional linc-mg ko mice. In addition to the muscle image shown here, I would also like to see the size of other muscles, in order to have a more thorough characterization of the role of linc-mg during myogenesis in vivo. Similar to the earlier comments, I would also like to see the size of both fast and slow muscle types (including the soleus muscle) in figure 3a.

Figure 4 shows an interesting increase in muscle size in the linc-mg transgenic mice. Similar to the previous comment, I would like to see the size of other muscles presented here. Also, the change in figure 4b is quite impressive but I am unable to see these differences in the myofiber CSA images. I would recommend cropping a smaller area of the H&E stained-slides in order to very clearly show an increase in CSA in the transgenic mice with fewer representative fibers.

Figure 5 provides important mechanisms for linc-mg function. However, according to the literature, similar types of analyses were also done for linc-MD1. Can the authors explain

why this mechanism is interesting in the context of lnc-mg? It would appear from the literature that other lncRNAs (such as linc-MD1) act in a similar manner. Would this be a more generalized mechanism of lncRNA action?

Figure 6 shows interesting results, however the changes in the protein levels are not very large. If lnc-mg is a major regulator of myogenesis, I would expect changes in Igf2 to be greater. The authors should comment on whether they believe that Igf2 is a major molecule involved in myogenesis downstream of lnc-mg. If so, what is their explanation for the small changes in Igf-2 in this figure? If not, do they believe that other molecules may show greater changes? I would recommend also examining the levels of other molecules that could be involved in regulation of myogenesis (not only Igf2) in this context downstream of lnc-mg.

Figure 7 shows a nice model of how lnc-mg may act during myogenesis. I believe this type of model can be very useful in publication. However, I believe this model can be improved in order to make it more visually appealing. For the left side schematic, I recommend adding a bit more color - for example, I would make each mouse a different color to distinguish between them. I would also show an actual skeletal muscle (from the various figures in the manuscript) instead of a cartoon for a more realistic visual of muscle size. Finally, in order to make the point that the myofiber CSA differs in the three types of mice, I would consider showing fewer muscle fibers in cross-section to make this point. For the right side, I would improve the schematic to show how the sponging of lnc-mg affects the actual process of myogenesis. Currently, I don't believe that the current schematic on the right side is very informative to the readers without putting it in the context of myogenesis.

Reviewer #2 (Remarks to the Author):

RE: NCOMMS-16-16008-T

The manuscript described the identification and characterization of a novel skeletal muscle-enriched lncRNA, lnc-mg. Using shRNA and overexpression studies in cultured myogenic cells the authors showed that this RNA functions to promote myogenesis. The authors also created muscle-specific ko and overexpression transgenic mouse models and demonstrated that the deletion mice developed atrophic muscle while the transgenic mice showed hypertrophic muscle. They provided evidence that lnc-mg contains miR-125b binding sites and they interact to regulate the expression of Igf2, a growth factor known to play important roles in muscle development.

The manuscript is very well written, and the data in generally support the conclusion. The authors did not find a human homolog of lnc-mg, but they did in the pig and sheep. Thus their findings have the potential value in farm animals.

Specific points:

While the effects of lnc-mg deletion and overexpression on muscle development are impressive, the mechanistic part is relatively weak. Especially, that "lnc-mg acts as a molecular sponge through antagonizing miR-125b to control Igf2 protein level during myogenesis" is an over statement which is not fully supported by the data. The authors did not specify the relative expression levels between lnc-mg and miR-125b in the cells. One would expect a much higher level of expression of a lncRNA relative to its target miRNA. Also, the authors need to show the subcellular localization of lnc-mg. Does it predominantly localize to the cytoplasm where one would expect it to interact with its target miRNA? The pull down assays (Fig. 5J) showed ~3-fold enrichment of lnc-mg in the biotin-miR125b. However, this did not tell us how big a fraction of the endogenous pool of lnc-mg was interacting with the miRNA. Being a "sponge", one would expect a robust interaction.

Supplementary Fig. 2d: lnc-mg overexpression is 600-1200 fold of the endogenous. This is a non-physiological level. I suspect it was a PCR artifact - DNA contamination from the transfected lnc-mg-expressing plasmid. The authors need to fix this problem.

The authors need to mention the age and sex of the animals used in the experiments. Some genes can have age/sex-specific effects.

Reviewer #3 (Remarks to the Author):

General/overarching comments to the author:

This is a comprehensive piece of work, novel and highly interesting. In its present form, the authors need to thoroughly proof read the manuscript. The abstract and introduction, in particular, are littered with inconsistencies. There are also parts of the methods, results, figures and figure legends that require more detail or clarification in order to confirm the studies robustness, as detailed in my specific comments below. The discussion also fails to highlight some of the most important conclusions.

Specific Comments:

Abstract

Authors state: 'Recent studies have highlighted important roles for long noncoding RNAs (lncRNAs) as essential regulators during tissue/organ development.' The authors go on to investigate adult myogenesis in the manuscript, so seems strange to open the abstract to include 'development,' rather than, for example, reference to repair and regeneration of adult skeletal muscle.

Authors include: 'Recent studies have highlighted important roles for a few long noncoding RNAs (lncRNAs)...' Remove 'a few.'

Authors state: 'explaining the role of lncRNAs in myogenic differentiation of stem cells and myogenesis in vivo.' Revise to: explaining the role of lncRNAs in myogenic differentiation of adult skeletal muscle stem cells (MuSCs) and myogenesis in vivo.

Poor sentence structure, authors write: 'In vivo, skeletal muscle specific knockout of mouse lnc-mg develops muscle atrophy and losing muscular endurance during exercise, while skeletal muscle-specific lnc-mg transgenic mice promotes muscle hypertrophy.'

Consider revising to: In vivo, skeletal muscle specific knockout of lnc-mg resulted in muscle atrophy and the loss of muscular endurance during exercise in mice. Alternatively, skeletal muscle-specific transgenic overexpression of lnc-mg promoted muscle hypertrophy in mice.

Authors mention 'In vitro analyses...' this needs expanding to include details of the cells derived. E.g. in-vitro analysis of primary skeletal muscle cells isolated from mice....

Authors include: 'Showed that lnc-mg is induced during myogenesis.' Details are required here e.g. lnc-mg is increased exponentially over the time course (0-5days) of myogenesis. Or is increased upto 3day and then plateaus at 5 days? Reviewer is unaware of the direction and the former sentence is hypothetical as I am still to read the results; however, more detail is required.

Authors state: 'and accelerates muscle stem cells differentiation in vitro..' More detail required e.g. improves muscle cell differentiation in-vitro when overexpressed?

Author uses terminology: 'Molecular sponge' then goes on to say 'through antagonizing...' I

am not sure this terminology holds up within the context of the sentence as a sponge is something that absorbs or perhaps reduces but does not antagonize...

The use of antagonize in this context is also not familiar as its definition is to cause (usually someone/something) to become hostile. As opposed to an antagonist in biochemistry that is used to describe an activator of something. Consider revising this sentence. By suggesting, e.g. lnc-mg results in increased miR-125b that subsequently controls protein abundance of Igf2.

Authors state: 'miR-125b to control Igf2 protein level during myogenesis...' A little more detail of how it does this and how it was demonstrated in the present study is required.

Authors include: 'development and the mechanisms that how lncRNAs coordinate the thickness of muscle fiber and muscular strength...' This is poorly written, consider using: 'mechanisms by which lncRNAs coordinate the size of skeletal muscle fibers and strength.'

Authors state: 'These findings identify lnc-mg as an important noncoding regulator for skeletal muscle.' Consider revising to: These findings identify lnc-mg as a novel and important noncoding regulator for skeletal muscle.

Introduction

The authors should make sure they clearly distinguish that they are introducing ADULT myogenesis and not embryonic myogenesis in the opening paragraph. As this is what they go onto present data for in the present study. Reading this, preceding the methods/results sections, makes the reader jump to the results to confirm the model used, so it should be stated up front.

Authors state: 'and then proceeds with cell proliferation and fusion.' Important to include migration is this sentence to read: ..and then proceeds with cell proliferation, migration and fusion.

Authors include: 'terminal differentiation is initiated to form multinucleated myotubes with contractile ability.' Consider revising to: terminal differentiation is initiated to form multinucleated myotubes with the capacity to contract.

Roles as signaling molecule, should read: Roles as signaling molecules

Authors state: 'Recently, a large class of lncRNAs, referred to as competing endogenous RNAs (ceRNAs), has been characterized.' This sentence requires a primary reference. Authors state: 'ceRNAs protect mRNAs by acting as sponges for microRNAs,' the papers the authors cites refer to these first as 'molecular sponges' so consider revising to: ceRNAs protect mRNAs by acting as 'molecular sponges' for microRNAs.

Authors state: 'Additionally, H19 has been demonstrated to act as a molecular sponge for let-7 to control skeletal muscle differentiation.' This should be replaced with allowing to read: Additionally, H19 has been demonstrated to act as a molecular sponge allowing let-7 to control skeletal muscle differentiation.

Immediately following the penultimate paragraph (prior to the last paragraph) in the

introduction, the authors must to include their aims and specific hypotheses for the study that stems from the extensive rationale for the study the authors include.

Authors mention: 'we describe the function of a new lncRNA, named lnc-mg.' How was this name decided? What does 'mg' stand for. This detail is required if this is a newly defined lncRNA. This is also the first time you refer to this in the main text so its full name (not just acronym) should be used here.

Methods

In Supplementary figure 1a the authors use the acroynms GM or DM to describe their growth and differentiation media respectively. However, these acroynms are not specifically described in the methods or figure legend (reviewer suggests to include this information in the methods section, under; 'In vitro cell culture and differentiation' authors should change wording from: 'DMEM with 10% FBS and induced to myogenic differentiation by switching to DMEM containing 2% horse serum.' To: DMEM with 10% FBS (growth media/GM) and induced to myogenic differentiation by switching to DMEM containing 2% horse serum (differentiation media/DM).

Under the real-time PCR section author's state in accordance with the MIQE quidelines and state a PMID, please include a in text citation/reference in line with the journals format.

Was there a set volume or concentration of cDNA loaded into the PCR reactions, please include information.

No detailed information is included on the PCR cycling protocol or number of cycles performed, this is required.

The authors require more information to ensure appropriate PCR conditions were met. Please include information in the methods on the stability and therefore the suitability of your chosen reference gene (Gapdh) across all experimental conditions. E.g. mean \pm Sd and % variation in the reference gene across conditions. If variation was low was a pooled reference ct value used in the delta delta ct equation or was the specific samples reference gene Ct value used.

How did you confirm that your PCR was specific to the gene of interest e.g. did you run at least a melt curve analysis that suggested a single product for each gene investigated. For example, upon performing BLAST searches for your myogenin primers taken from Macpherson et al., (Cell Biochem 2011), they amplify a product of 178 bp on the myogenin gene.

However, it also shows potential unintended target of the Fem1b gene of 331bp, although a little large for real time PCR, a melt analysis would at least confirm that one single product was amplified.

Please include mean, SD and variation for the efficiency of PCR reactions across all conditions. If there was large variation e.g. above 10% was the analysis tailored to account for this e.g. use of REST software etc.

What was used as the calibrator condition in the relative delta delta PCR analysis. Include this in the methods or in the figure legends where the PCR data is included.

In table 1 - PCR primers, please include the product lengths (bp) for each gene e.g. a blast of your myoD primers, these amplify a product of 121bp. Myogenin 178bp etc.

In table 1 please include the accession number for each of the genes e.g. myoD is NM_010866.2.

Cell Transfection section: When the author uses a weight of e.g. Inc-mg expression plasmid (500 ng). Is this the final total amount in each 24 well plate or is this 500ng/ml of media with 0.5 or 1 ml of media used per well? Please make this clear.

Also under 'cell transfection', you need to include a reference to the method of confirming Inc-mg silencing/overexpression. Expect this was PCR but authors need to state this, what they did and perhaps refer to the specific figure legends where the silencing/overexpression are demonstrated.

Under 'immunofluorescent staining'. Your antibody for MyHC detects only adult type IIx myHCs. If staining primary cells while fusing or when differentiated you need to justify why you used this particular myHC at the timepoints the cells/myotubes were fixed and stained for myHC IIx.

Under. 'Exercise performance test': Authors state: Briefly, mice were first accustomed to treadmill running on a 20{degree sign} incline and 25 cm/s belt speed for 3 days. Did the mice run on treadmills for 3 days without rest? How long did they exercise per day over the three day period. Was this 20 minutes like on the 4th day of exercise as specified in the next sentence?

Under section Anti-Ago2 immunoprecipitation, author requires a SPACE after 25mM Tris-Cl. Also, where author's state: 'magnetic beads was added' should read, magnetic beads were added. At the end of this section the author's state: The RNA was extracted from the remaining beads with TRIzol Reagent (Life Technologies) and evaluated by real-time PCR assay. What exactly was evaluated by PCR and what was the specific protocol, if different or if the same as previous PCR methods then refer back to the methods above.

Under section: 'Biotin-labeled miR-125b capture,' the final sentence you mention, once biotin labeled, RNA was extracted and PCR assay performed. What were specific PCR requirements for this particular analysis, information is required.

The authors also perform microarrays for miRNA's prior to the above reporter assay. There is no detail of these microarray methods in the methods text.

There is nothing in the methods as to how the authors measured muscle weight and when (at what time point) animals were sacrificed for this. Also, measurement of force and specific force of the tissue is not detailed in the methods.

Histological analysis of muscle sections stained with hematoxylin and eosin (Figure 3b) is also not detailed. This information is required in the methods.

ELISA analysis, what were the excite/emission spectra you detected at.

Results

Authors state in the results text under the, 'Inc-mg is induced in myogenesis' section: 'Microarray data from quiescent and' how do the authors know these were quiescent versus actively cycling. In Fig 1a, the microarray data shows the panel on the left as GM, so these cells were in growth media 10% serum. A typical skeletal muscle cell quiescent media contains only 0.1% serum. Do the authors simply mean undifferentiated muscle cells?

As suggested above, there is no text in the methods for microarray protocols.

Author's state under 'Inc-mg is induced in myogenesis' ..and enriched in skeletal muscle.' This should read that was enriched in skeletal muscle.

The authors briefly refer to figure 1c, but the text describes fig1b and there is no description of the data in fig1c on the expression of Inc-mg between muscle groups that show some quite different expression values between muscles.

In supplementary figure 1, figure legend the authors state: 'Characterization of Inc-mg. (a) MyHC immunostaining of undifferentiated muscle stem cells (DM) and 5 days of differentiated muscle stem cells (GM).' This does not make complete sense as GM is used for cell proliferation not differentiation and DM media is used for differentiation and should not therefore be linked with undifferentiated muscle stem cells as the description suggests. I think the DM and GM in brackets need swapping around.

In the figure legend for figure 1. Authors state: '(skeletal muscle value was normalized to 1.0).' As suggested in the reviewers methods comments above the appropriate calibrator condition in the relative gene expression analysis should be referred to here.

Furthermore, in the authors supplementary figure 1 the authors state: 'muscle stem cells were treated with Calf Intestine Alkaline Phosphatase (CIP) to remove free 5' -P, then treated with Tobacco Acid Pyrophosphatase (TAP) to remove the cap structure and then a RNA adapter oligonucleotide was ligated to the RNA population using T4 RNA ligase (FistChoice RLM-RACE Kit, Ambion). (f) Bioinformatics analysis of the coding capability of Myh1, H19 and Inc-mg.' These methodologies are not included in the method sections. I am not sure what journal guidelines say about data shown in supplementary figures, but I still think all the methods should be present in the main methods section as the sup figure 1 is referred to in the main results text.

Phase/contrast images in figure 2 are really poor quality; the cells/myotubes can hardly be distinguished. The authors should replace with better quality images or remove them and simply keep the fluorescent images only.

In the final sentence in section 'Inc-mg promotes MuSCs differentiation', authors need to remove extra space and replace comma where it reads: In addition, Inc-mg

The reviewer thinks that supplementary figure 2 should not be a supplementary file but a figure for the main manuscript, as this is key for the generation of the following/ later data.

Under, Conditional knockdown of Inc-mg in skeletal muscle mice results in muscle atrophy and weakness in vivo, authors state: 'Moreover, the muscle fiber in gastrocnemius cross sections was thinner in Inc-mgskl-/- mice (Fig. 3b,c). Measurement of the mean diameter of muscle fibers revealed that Inc-mgskl-/- mice had more thin fibers (Fig. 3d).' Again there is no reference to how these parameters were measured in the methods section.' Also, this is poorly written consider revising to: Moreover, the cross-sectional area of muscle fibers in the gastrocnemius were smaller in Inc-mgskl-/- mice (Fig. 3b,c). Measurement of the mean diameter of muscle fibers revealed that Inc-mgskl-/- mice had a larger number of thinner fibers (Fig. 3d).

There is no mention of specific force in the results text (but you include a figure under fig 3e.) in the results text and the differences in control vs. Inc-mgskl-/-

Figure 3: Why are the CSA's relative to control and not absolute e.g. mm²

Figure 3: (d) Distribution and mean diameter of muscle fibers in gastrocnemius. How is the y axis for 'ratio' calculated, why is this not simply a frequency or number of fibers that fall within each category of fiber diameter.

Figure 4g. Does denervation induce a loss of muscle in the control group as you would expect. I.e. are the white bar controls vs. denervated in WT significantly different. If they were not, then authors need to explain why this is the case.

In figure 4g. What are the percent changes in control WT vs. TG and denervated WT vs. TG. Are they comparable or different, if different are they significantly different? It looks like control increases from 100% in WT to 160% in TG (approx. 60% increase) and in the denervated groups CSA increases from 75% in WT to 125% in TG (approx. a 50% increase). Therefore, as questioned above, is this increase significantly different or not? E.g. it is quite extraordinary that on a background of denervation that the response/magnitude of increased muscle size to overexpression of Inc-mg is similar (50-60%) to that of WT. Authors should discuss this in more detail in the discussion as this is an intriguing finding and relevant for age-related muscle loss where we know denervation occurs.

Figure 6 includes real time pcr analysis of miR125b. What are the primer/probe details and specific pcr requirements.

Discussion

First sentence: 'Function as sponges,' revise to, function as molecular sponges.

Authors state:..'resulting in increased Igf2 protein to enhance myogenesis.' Quthors should include original work on Igf2 improving differentiation e.g. Stewart et al., 1996 PMID: 8841419 and Stewart and Rotwein 1996, PMID: 8626686

Authors require a comma after: 'and mechanistic characterization of Inc-mg, both...'

The final paragraph in the discussion: 'Although Inc-mg homolog in humans has not yet been found, we cloned potential homologous IncRNAs from pig and sheep. Because skeletal muscle-specific Inc-mg transgenic mice had much bigger and stronger muscle compared to their littermates, we further generated muscle-specific Inc-mg homologous IncRNA transgenic sheep. These sheep were much bigger with enhanced muscularity compared to wild type lambs (data not shown). Consequently, in addition to providing an interesting new mechanism that regulates myogenesis in several mammalian species, Inc-mg and its homologous genes could serve as useful candidates to improve muscle production in farm animals.'

This is highly speculative and not a conclusion based on the data presented. Unless authors plan to include the lamb data they refer to then conclusions should be made within the context of the data presented.

The authors should focus on the interesting findings (as suggested in the results comments above) on the rescue of muscle size following denervation in transgenic animals for Inc-mg as a key finding.

Reviewed by:

Adam P. Sharples BSc (Hons) MSc. FHEA. PhD.

Senior Lecturer in Cellular and Molecular Physiology

Stem Cells, Ageing and Molecular Physiology Unit

Exercise Metabolism and Adaptation Research Group

Research Institute for Sport and Exercise Sciences

Editor-in-Chief (Cellular and Molecular Exercise Physiology)

Life Sciences Building (rm 1.17)

Byrom Street, Liverpool, UK, L3 3AF

Twitter: @DrAdamPSharples / @CelMolExPhysiol

Profile and Publications: <https://www.ljmu.ac.uk/about-us/staff-profiles/faculty-of-science/sport-and-exercise-sciences/adam-sharples>

Response to the comments from Reviewer 1 (NCOMMS-16-16008-T)

The authors highlight an important, novel, skeletal muscle-enriched lncRNA (named lnc-mg) and its functions and mechanisms during myogenesis. This work has the potential to open up a new area of investigation on lnc-mg, which will also likely be studied in multiple contexts in future studies. Below, I provide a few general comments, as well as comments about specific figures, which I think will improve this manuscript.

Answer (A): Thank you for reviewer's comments. The comments or suggestions raised by the reviewer are very constructive and very helpful for revising and improving this manuscript.

General comments:

While the abstract highlights the major, important findings related to lnc-mg in myogenesis, I found the statement that "experiments showing the role of lncRNAs in myogenic differentiation of stem cells and myogenesis *in vivo* haven't been done" (lines 53-55) to be inaccurate.

A: Thanks for the reviewer's comments. We agree with the reviewer that the statement "experiments showing the role of lncRNAs in myogenic differentiation of stem cells and myogenesis *in vivo* haven't been done" is inaccurate. It was replaced by "*in vivo*, the role and mechanism of lncRNAs in myogenic differentiation of adult skeletal muscle stem cells (MuSCs) and myogenesis are still largely unknown." in the updated manuscript. Please refer to **lines 55-57** in revised manuscript.

From my knowledge of the current literature, several lncRNAs have been studied in the context of myogenic differentiation and the majority was shown to promote myogenesis.

A: Thanks for the reviewer's concerns. It is true as reviewer suggested that there are several lncRNAs involved in myogenic differentiation, such as lnc-MD1 (Cesana, M. et al. 2011), H19 (Kallen, A.N. et al. 2013), Malat1 (Han, X.R. et al. 2015), MUNC (Mueller, A.C.

et al. 2015), LncMyoD (Gong, C.G. et al. 2015), Inc-YY1 (Zhou, L. et al. 2015) and lncRNA-Dum (Wang, L.J. et al. 2015), most of which were shown to promote myogenesis, such as Inc-MD1 (Cesana, M. et al. 2011), MUNC (Mueller, A.C. et al. 2015), LncMyoD (Gong, C.G. et al. 2015), Malat1 (Han, X.R. et al. 2015) and Inc-YY1 (Zhou, L. et al. 2015). We had cited these references in the introduction part of revised manuscript. Please refer to **lines 80-92** in revised manuscript.

Moreover several of them are also involved in muscle regeneration (therefore *in vivo* studies have been done).

A: As the reviewer pointed, several lncRNAs, which involved in myogenesis, have been reported in muscle regeneration *in vivo*. For instance, LncMyoD (Gong, C.G. et al. 2015) and Inc-YY1 (Zhou, L. et al. 2015) were induced during mice muscle regeneration which we mentioned in the revised manuscript. In addition, we also detected the expression of lnc-mg during mice muscle regeneration. We employed a widely used muscle regeneration model in which the injection of cardiotoxin (CTX) resulted in muscle injury and in turn induced muscle regeneration. It was found that the expression of lnc-mg was induced during muscle regeneration, and the peak of expression level was around day 2 to day 3 (Please refer to **Figure** below).

Real-time PCR analysis of the expression of lnc-mg during CTX-induced regeneration. Mean values \pm SEM, n=4. Mice were 8-week old, two for male, two for female

Similarly, in the introduction, the authors state that ceRNA studies in mice are novel (lines 88-89; 93-94), which is not entire true given that ceRNAs have been studied *in vivo*. I recommend the authors revise these statements to less strong statements and instead highlight how their findings on Inc-mg are novel to myogenesis both *in vitro* and *in vivo*.

A: Thanks for the reviewer's comments. It is true as reviewer suggested that some ceRNAs have been studied *in vivo* (for example, Karreth, F. A. *et al.* 2015). Thus we have revised the statement "While functions of these lncRNAs have been identified *in vitro*, their *in vivo* roles have not been established." to "While partial functions of these lncRNAs have been identified *in vitro* or even preliminary *in vivo*, most of their roles for myogenesis are still waiting for disclosing." (Please refer to **lines 101-102** in revised manuscript). And we revised the sentence "Our study identified an effectively ceRNA that controls myogenesis in mice." of the last version to "Our study further reveals that Inc-mg could promote myogenesis and enhance muscle mass *in vivo*." (Please refer to **lines 107-108** in revised manuscript).

I would recommend the authors become familiar with the literature of lncRNAs in myogenesis and cite relevant papers. Showing the readers that the authors are familiar with the latest literature on the topic, as well we showing the major trends known about lncRNAs in myogenesis (therefore broadening the introduction) would be important changes to make in order to create a context in which the novelty of Inc-mg could be better highlighted. Finally, for a more comprehensive analysis of lncRNAs in myogenesis, please see the review article Simionescu-Bankston and Kumar, JMM, 2016 (as well as other excellent published reviews on non-coding RNAs in myogenesis).

A: Thanks for the reviewer's suggestions. We appreciate for the review article recommended by the reviewer. Firstly, we have referred to the latest literatures about lncRNAs which involved in myogenesis and re-written the related part in the introduction. Please refer to revised statement below: "It is worth noting that some lncRNAs have been determined to regulate myogenesis (Simionescu-Bankston, A. & Kumar, A. 2016,

Neguembor, M. V. *et al.* 2014). For example, ncRNA SRA was reported to promote myogenic differentiation by regulating the transcriptional activity of MyoD (Caretto, G. *et al.* 2006 and Hube, F. *et al.* 2011). H19 has a critical role in skeletal muscle differentiation and regeneration which is mediated by miR-675-3p and miR-675-5p encoded within H19 (Dey, B.K. *et al.* 2014). MUNC, located upstream of MyoD and specifically expressed in skeletal muscle, is a lncRNA which can promote myogenesis by regulating MyoD expression (Mueller, A.C. *et al.* 2015). Similarly, lncMyoD, activated by MyoD, plays an important role in promoting myogenesis and skeletal muscle regeneration (Gong, C.G. *et al.* 2015). lnc-MD1 (Cesana, M. *et al.* 2011), Glt2/Meg3 (Neguembor, M.V. *et al.* 2014), lnc-YY1 (Zhou, L. *et al.* 2015) and lncRNA-Dum (Wang, L.J. *et al.* 2015) are also believed as important positively regulators of myogenesis. In contrast, recent studies have shown that certain lncRNAs negatively regulate myogenesis. For instance, m^{1/2}-sbsRNA inhibits myogenesis via reducing TRAF6 by Staufen-mediated mRNA decay (Wang, J. *et al.* 2013). Yam (YY1-associated muscle lncRNAs) (Lu, L.N. *et al.* 2013), H19 (Kallen, A.N. *et al.* 2013), lnc-31 (Ballarino, M. *et al.* 2015) and Sirt1 AS lncRNA (Wang, J.S. *et al.* 2013 and Wang, G.Q. *et al.* 2016) were reported to inhibit myogenic differentiation.” (Please refer to **lines 80-92** in revised manuscript).

Specific comments:

Other than the general comments above, many of the differences shown in the manuscript are not major, which makes it difficult to argue that lnc-mg is a critical regulator of myogenesis. One would expect greater differences in levels of lnc-mg in myogenesis *in vitro* and *in vivo* if this were the case. Below, I provide specific examples/comments for particular figures. The authors are advised to take these comments into consideration and fix the issues presented here in all places in the manuscript (even if they are only mentioned in one place in this critique):

A: Thanks for the reviewer’s comments and the suggestions were well taken.

In Figure 1b, the increased Inc-mg expression in skeletal muscle is not very large (same comment for figure 2c), posing the question of whether Inc-mg plays a major role here. Nevertheless, the error bars are small and the increase in expression in skeletal muscle appears to be statistically significant visually. The authors need to discuss the statistics utilized to obtain these data, and whether the increase in skeletal muscle is statistically significant.

A: Thanks for the reviewer's comments. Compared to other control tissues, the expression level of Inc-mg in skeletal muscle was higher but not obvious in the original **Fig. 1b**. The reason lies the "y-axis" of original **Fig. 1b** represents \log_{10} value based on RNA sequence data which was mislabeled to real-time PCR in figure legend of last version. To show the results in a more reliable way and to highlight the elevated expression of Inc-mg in skeletal muscle, we provided the data by real-time PCR analysis in revised **Fig. 1b**. It was found that Inc-mg expression in skeletal muscle was much higher than other tissues with statistical significant difference assessed by the Student's t-test. (Please refer to **Fig. 1b** in revised manuscript).

Given that this manuscript focuses on Inc-mg in skeletal muscle, the authors also need to explain why they examined the levels of Inc-mg in so many tissues. I would consider showing a few major tissues here for comparison (for example skeletal muscle, heart, kidney, liver, lung) and have the rest of the data as supplementary.

A: Thanks for the reviewer's comments. The main purpose of examining the levels of Inc-mg in many tissues is that we want to validate whether Inc-mg is highly expressed in skeletal muscle. We agreed with the reviewer's suggestions and compared the data of heart, liver, spleen, lung and kidney with skeletal muscle in the updated **Fig. 1b**. In addition, we presented the rest data of Inc-mg expression in other tissues as **Supplementary Fig. 1b**. Please refer to **Fig. 1b** and **Supplementary Fig. 1b** in revised manuscript.

In Figure 1c, the comments are similar to the ones above. It is currently unclear from the presented data why the expression was analyzed in so many different

types of muscles. This is also not explained thoroughly in the text referring to this section of figure 1. Similarly to 1b, the authors could show data in only a few muscles here and the rest be presented as supplementary data. These changes would make the manuscript more focused on the important findings on Inc-mg in myogenesis.

A: Thanks for the reviewer's comments. To test whether Inc-mg differently expressed in various types of muscles, we examined the levels of Inc-mg in 15 types of skeletal muscles which located in different places. To make it clear, we presented the data of extensor digitorum longus (EDL), tibial anterior (TA), soleus muscle (SOL) and gastrocnemius muscle (GAS) in revised manuscript according to the reviewer's suggestion (Please refer to **Fig. 1c**). And we placed the data of other muscles in **Supplementary Fig. 1c** and explained thoroughly in the text **lines 119-122** to make the manuscript more focused on the role of Inc-mg in myogenesis. Please refer to **Fig. 1c** and **Supplementary Fig. 1c** and **lines 119-122** in revised manuscript.

The authors also need to explain the meaning behind the data presented in Figure 1c for muscle biology in general. For example, are all the muscles examined here containing the fast-fiber type, and if so, what does it mean that the expression increases in these muscles? Since Inc-mg is a novel regulator of myogenesis, a thorough analysis of its expression in both slow and fast muscle types (including TA and soleus) would be beneficial to the muscle field.

A: Thanks for the reviewer's comments. We explored the expression profile of Inc-mg in different types of muscles which contain fast-fiber type, slow-fiber type and mixed-fiber type. It suggested that there is a little difference about Inc-mg expression among tested muscles (**Fig. 1c** and **Supplementary Fig. 1c**). This interesting founding need to be investigated in future research. However, the main purpose of this study is to validate whether Inc-mg expression in muscles is obviously higher than other tissues. Thus, we present the Inc-mg expression data of EDL, TA, SOL and GAS in revised **Fig.1c**, which is included both fast and slow fibers. Please refer to **Fig. 1c** and **Supplementary Fig. 1c** in revised manuscript.

With regards to Figure 1d, I believe that generally data showing changes using primary muscle cells is much more relevant and valuable than C2C12 data. In addition, in this case, the data in the two cell types looks exactly the same; therefore I do not believe the C2C12 data add much information. Unless the authors have a very compelling reason for showing these data in both cell types, I recommend removing the C2C12 data.

A: Thanks for the reviewer's comment. For the data shown in original Fig. 1d, the variation tendency of Inc-mg in C2C12 cells is similar to primary muscle cells, thus we removed the C2C12 data in the revised manuscript. Please refer to Fig. 1d in revised manuscript.

In Figure 2b, both the phase and fluorescent images of myotubes (both here and everywhere else in the manuscript) are of very poor quality. One cannot discern the myotubes in the phase images, and the number of nuclei in the fluorescent images is very hard to determine. I would have a hard time believing graphs based on these current images. Therefore, very high quality images of muscle cells/myotubes are necessary. In addition to high quality images, a more thorough analysis of myotube size is needed - for example how many myotubes with different numbers of nuclei there are etc. I would also like to see a graph showing indicating that the same cell number was plated/used - showing this information would give confidence that smaller myotubes observed at the end of the experiment are not due to fewer cells being plated in the beginning of the experiment. This analysis is critical to concluding that Inc-mg has a role in myogenesis and specifically myotube formation.

A: Thanks for the reviewer's comments and the suggestions were well taken. We redo the MyHC staining experiments and take photos with better quality. We displayed the phase photo as T-Figure 1 and replaced fluorescence image with better quality in revised Fig. 2b and Fig. 2f. Please refer to Fig. 2b and Fig. 2f in revised manuscript and T-Figure 1a and T-Figure 1b below. In addition, we added a graph in T-Figure 1c below indicating that the same cell number ($\sim 10^4$) was plated in 24-well plate in the beginning of the

experiment. This data suggested that smaller myotubes observed at the end of the experiment were not due to fewer cells being plated but due to the Inc-mg effect on myotubes formation.

T-Figure 1

MyHC immunostaining of MuSCs transfected with control shRNA or Inc-mg shRNA then cultured in DM for five days. Scale bar: 40 μ m.

MyHC immunostaining of MuSCs transfected with control vector or Inc-mg vector then cultured in DM for five days. Scale bar: 40 μ m.

10×10^4 MuSCs were seeded in 24-well plates for myogenic differentiation.

Figure 3a shows a very nice decrease in muscle size in the conditional Inc-mg ko mice.

A: We appreciated very much for the reviewer's kind comments.

In addition to the muscle image shown here, I would also like to see the size of other muscles, in order to have a more thorough characterization of the role of Inc-mg during myogenesis in vivo. Similar to the earlier comments, I would also like to see the size of both fast and slow muscle types (including the soleus muscle) in figure 3a.

A: Thanks for the reviewer's valuable suggestion. We agree with the reviewer's opinion of supplying the size changes of both fast and slow muscle types is helpful to show a more thorough characteristic of the role of Inc-mg during myogenesis *in vivo*. We have added the photo of dissected GAS, SOL, EDL and TA from Inc-mg skeletal muscle-conditional knockout mice and control loxp mice in **Fig. 3a**. Please refer to **Fig. 3a** in revised manuscript.

Figure 4 shows an interesting increase in muscle size in the Inc-mg transgenic mice.

A: We appreciated very much for the reviewer's comments.

Similar to the previous comment, I would like to see the size of other muscles presented here.

A: Thanks for the reviewer's valuable suggestion. Similar to the previous suggestion raised by reviewer, we have added the photo of desected GAS, SOL, EDL and TA from Inc-mg transgenic mice and wild-type mice in **Fig. 4a**. Please refer to **Fig. 4a** in revised manuscript.

Also, the change in figure 4b is quite impressive but I am unable to see these differences in the myofiber CSA images. I would recommend cropping a smaller area of the H&E stained-slides in order to very clearly show an increase in CSA in the transgenic mice with fewer representative fibers.

A: Thanks for the reviewer's comments. We are sorry for using unrepresentative image in original **Fig. 4b**. To show the clearly increase in CSA of transgenic mice, we have reprocessed the higher magnification H&E image in **Fig. 4b** with fewer representative fibers. Please refer to **Fig. 4b** in revised manuscript.

Figure 5 provides important mechanisms for Inc-mg function. However, according to the literature, similar types of analyses were also done for linc-MD1. Can the authors explain why this mechanism is interesting in the context of Inc-mg? It would appear from the literature that other lncRNAs (such as linc-MD1) act in a similar manner. Would this be a more generalized mechanism of lncRNA action?

A: Thanks for the reviewer's comments. We were interested about the lncRNA function as endogenous miRNA sponge, because of growing evidence published recently to support ceRNA hypothesis (Such as Qu, L. *et al.* 2016, Xu, C. *et al.* 2016 and Yan, B.A. *et al.* 2015). Also from the review of endogenous microRNA sponges: evidence and controversy (Thomson, D.W. *et al.* 2016), which discussed the evidence and controversy of the ceRNA hypothesis. Although linc-MD1 (Cesana, M. *et al.* 2011), H19 (Kallen, A.N. *et al.* 2013) and our data indicated that Inc-mg could be recognized as a ceRNA to regulate miR-125b during myogenesis, we still believe that further investigations are needed to provide sufficient evidences to support the possibility that ceRNA be generalized mechanism of

lncRNA action.

Figure 6 shows interesting results, however the changes in the protein levels are not very large. If lnc-mg is a major regulator of myogenesis, I would expect changes in Igf2 to be greater.

A: Thanks for the reviewer's comments. We agreed with the reviewer that the changes of Igf2 protein levels are not very large in Fig. 6. Thus, we re-examined the expression levels of Igf2 protein in recently prepared muscle samples. It was found that Igf2 protein was approximate 50% lower in muscle from cKO mice and about 2.5-fold higher in muscle from transgenic mice. In **Fig. 6b** and **Fig. 6e**, Igf2 protein was ~50% lower in cKO mice and ~ 2.5-fold higher in transgenic mice. Although the changes were not very large, data are still with statistical significant difference among groups. Please refer to **Fig. 6b** and **Fig. 6e** in revised manuscript.

The authors should comment on whether they believe that Igf2 is a major molecule involved in myogenesis downstream of lnc-mg. If so, what is their explanation for the small changes in Igf2 in this figure? If not, do they believe that other molecules may show greater changes? I would recommend also examining the levels of other molecules that could be involved in regulation of myogenesis (not only Igf2) in this context downstream of lnc-mg.

A: Thanks for the reviewer's comments. To our knowledge and based on our data, we believed that Igf2 is a key molecule involved in myogenesis downstream of lnc-mg. Because Igf2 is a secreted growth factor, we examined the Igf2 protein levels in mice serum. It is found that Igf2 is about 55% lower in serum from cKO mice (**Fig. 6c**) and about 2.3-fold higher in serum from transgenic mice (**Fig. 6f**). It is identical to the results of Western-blot. Moreover, we tested the phosphorylation levels of PI3K and PDK1, which are crucial downstream signaling molecules of Igf2, in muscle from cKO mice or transgenic mice. It is found that the phosphorylation of PI3K and PDK1 was also significantly changed (**T-Figure 2**). Please refer to **Fig. 6c** and **Fig. 6f** in revised manuscript and **T-Figure 2** below.

T-Figure 2

Western blot analysis of P-PI3K, PI3K, P-PDK and PDK in muscle from Inc-mg cKO mice and Inc-mg transgenic mice.

Figure 7 shows a nice model of how Inc-mg may act during myogenesis. I believe this type of model can be very useful in publication.

A: Thanks for the reviewer's comments.

However, I believe this model can be improved in order to make it more visually appealing. For the left side schematic, I recommend adding a bit more color - for example, I would make each mouse a different color to distinguish between them.

A: Thanks for the reviewer's suggestion. In order to easily distinguish and describe the black colored C57BL/6J mouse between groups in our study, we make marks of cKO, WT and TG to represent as Inc-mg conditional knockout mouse, Wide-type mouse and Inc-mg TG mouse separately. Please refer to **Fig. 7** and **T-Fig. a** below.

I would also show an actual skeletal muscle (from the various figures in the manuscript) instead of a cartoon for a more realistic visual of muscle size.

A: Thanks for the reviewer's suggestion. We tried to replace the cartoon muscle with the realistic visual muscle. However, judging from the apparent quality of the actual skeletal muscle images, we thought the cartoon muscle could reflect the mass changes of muscle better. Please refer to **Fig. 7** and **T-Fig 3** below.

Finally, in order to make the point that the myofiber CSA differs in the three types of mice, I would consider showing fewer muscle fibers in cross-section to make this point.

A: Thanks for the reviewer's suggestion and concern is well taken. We have modified the muscle picture with fewer muscle fibers in revised **Fig. 7** and **T-Fig 3** below.

For the right side, I would improve the schematic to show how the sponging of lnc-mg affects the actual process of myogenesis. Currently, I don't believe that the current schematic on the right side is very informative to the readers without putting it in the context of myogenesis.

A: Thanks for the reviewer's suggestion and concern is well taken. We have modified the picture to show how the sponging of lnc-mg affects myogenesis in revised **Fig. 7**. We hope this new version of cartoon will be better. Please refer to **Fig. 7** and **T-Fig 3** below.

Response to the comments from Reviewer 2 (NCOMMS-16-16008-T)

The manuscript described the identification and characterization of a novel skeletal muscle-enriched lncRNA, Inc-mg. Using shRNA and overexpression studies in cultured myogenic cells the authors showed that this RNA functions to promote myogenesis. The authors also created muscle-specific ko and overexpression transgenic mouse models and demonstrated that the deletion mice developed atrophic muscle while the transgenic mice showed hypertrophic muscle. They provided evidence that Inc-mg contains miR-125b binding sites and they interact to regulate the expression of *Igf2*, a growth factor known to play important roles in muscle development. The manuscript is very well written, and the data in generally support the conclusion. The authors did not find a human homolog of Inc-mg, but they did in the pig and sheep. Thus their findings have the potential value in farm animals.

A: Thanks for the reviewer's kind comments.

Specific points:

While the effects of Inc-mg deletion and overexpression on muscle development are impressive, the mechanistic part is relatively weak. Especially, that "Inc-mg acts as a molecular sponge through antagonizing miR-125b to control *Igf2* protein level during myogenesis" is an over statement which is not fully supported by the data.

A: Thanks for the reviewer's constructive and helpful comments. As the reviewer suggested "Inc-mg acts as a molecular sponge through antagonizing miR-125b to control *Igf2* protein level during myogenesis" was an over statement. We revised the statement to "Inc-mg promoted myogenesis, by functioning as a competing endogenous RNA (ceRNA) for miR-125b to control protein abundance of *Igf2*." (Please refer to **lines 63-65**). In addition, we added new supplementary data to support the mechanistic part. Firstly, overexpression of Inc-mg led to the increased enrichment of Ago2 on Inc-mg, while substantially decreased enrichment on *Igf2* (**T-Fig. 1a** below). Secondly, the luciferase activity of *Igf2* 3'UTR reporters was increased upon wild-type Inc-mg overexpression but

not upon miR-125b binding site mutated Inc-mg (**T-Fig. 1b** below). Finally, streptavidin capture analysis suggested that the binding enrichment of miR-125b on *Igf2* decreased with overexpression of wild-type Inc-mg, but not with miR-125b binding site mutated Inc-mg (**T-Fig. 1c** below). Collectively, these new evidences in **T-Fig. 1** below and **Fig. 5** in last version demonstrated that Inc-mg could elevate *Igf2* expression through competing for miR-125b. Please refer to **T-Fig. 1a, b, c** below or **Supplementary Fig. 4, Fig. 5** and **lines 63-65** in revised manuscript.

T-Figure 1

a.

RIP assay of the enrichment of Ago2 on Inc-mg, *Igf2* and *Gapdh* relative to IgG in C2C12 cells transfected with Inc-mg empty vector (control vector) or Inc-mg overexpression vector (Inc-mg). Mean values \pm SEM, n=4, * $P < 0.05$.

b.

The relative luciferase activity of psiCHECK-2 containing *Igf2* 3'UTR co-transfected with miR-125b, miR-125b and Inc-mg, miR-125b and mutated Inc-mg respectively. Mean

values \pm SEM, n=4, * P < 0.05.

c.

Streptavidin capture was performed for C2C12 myoblasts co-transfected with bio-miR-NC or bio-miR-125b and *Igf2* 3'UTR, *Igf2* 3'UTR with Inc-mg or *Igf2* 3'UTR with Inc-mg-mut, followed by real-time PCR to detect *Igf2* mRNA and *Gapdh* mRNA levels. Mean values \pm SEM, n=4, * P < 0.05.

The authors did not specify the relative expression levels between Inc-mg and miR-125b in the cells. One would expect a much higher level of expression of a lncRNA relative to its target miRNA.

A: Thanks for the reviewer's constructive and helpful suggestions. To specify the relationship between Inc-mg and miR-125b expression levels in cells, we transfected different concentrations of Inc-mg overexpression vector into C2C12 cells. As results shown, miR-125b expression decreased upon the overexpression of Inc-mg in a dose depended manner (**T-Fig. 2a**). Then we detected the endogenous miR-125b expression in differentiating C2C12 cells and found that the expression of miR-125b was negatively related with Inc-mg expression (**T-Fig. 2b**). To make our conclusion more accurate, we combined **T-Fig. 1** and **T-Fig. 2** as **Supplementary Fig. 4** in the revised manuscript.

T-Figure 2

a.

Real-time PCR analysis of the expression of Inc-mg and miR-125b in C2C12 cells after transfected with 200 ng, 400 ng or 600 ng Inc-mg expression vector for 48 h. Mean values \pm SEM, n=4.

b.

Real-time PCR analysis of the relative expression of Inc-mg and miR-125b during C2C12 cells myogenic differentiation. Mean values \pm SEM, n=4.

Also, the authors need to show the subcellular localization of Inc-mg. Does it predominantly localize to the cytoplasm where one would expect it to interact with its target miRNA?

A: Thanks for the reviewer's constructive and helpful comments. To understand how

lnc-mg regulates myogenesis, we examined its localization by using single molecule RNA fluorescent *in situ* hybridization (FISH) in C2C12 cells and found that lnc-mg located both in cytoplasm and in nucleus (**T-Fig. 3a**). Then we performed this experiment upon sub-cellular fractionation of C2C12 cells, actin and malat1 (a nuclear lncRNA) were set as controls. As expected, the actin transcripts were localized to the cytosol, while malat1 was confined localized to the nucleus. Levels of lnc-mg analyzed by real-time PCR showed that lnc-mg located both in cytoplasm and nucleus (**T-Fig. 3b**). Additionally, levels of lnc-mg in cytoplasm of undifferentiated C2C12 cells and four days differentiated C2C12 cells were detected, and the amount of lnc-mg in cytoplasm showed a significant increase compared to undifferentiated cells (**T-Fig. 3c**). These data indicated that lnc-mg in cytoplasm may has important function upon differentiation. To make our data more convincing, we add **T-Fig. 3** as **Supplementary Fig. 3** in the revised manuscript.

T-Figure 3

The subcellular localization of lnc-mg in C2C12 cells determined by fluorescent *in situ* hybridization (FISH). Scale bar: 20 μ m.

Real-time PCR analysis of the subcellular localization of lnc-mg in C2C12 cells. Mean

values \pm SEM, n=4.

Real-time PCR analysis of Inc-mg level in cytoplasm of undifferentiated C2C12 cells and four days differentiated C2C12 cells. Mean values \pm SEM, n=4, * $P < 0.05$.

The pull down assays (Fig. 5J) showed ~3-fold enrichment of Inc-mg in the biotin-miR125b. However, this did not tell us how big a fraction of the endogenous pool of Inc-mg was interacting with the miRNA. Being a "sponge", one would expect a robust interaction.

A: Thanks for the reviewer's comments. To our knowledge, **Fig. 5J** is a streptavidin capture but not a pull down assay. It's a half-endogenous assay, because we transfected only bio-miR-125b but not Inc-mg into cells. So the Inc-mg captured by miR-125b was endogenous. Meanwhile, we referred to the methods from literatures (Qu, L. *et al.* 2016 and Kallen, A.N. *et al.* 2013) about how to detect endogenous lncRNAs interacting with miRNAs but found that there were still lacking good analysis tools for testing the endogenous pool of lncRNA interacting with the endogenous miRNA up to date. Because of the limitations of research methods for ceRNA, it is difficult to quantify the endogenous interaction among lncRNA, miRNA and targets (reviewed by Thomson, D.W. *et al.* 2016). In addition, we added the supplementary data supporting the statement that Inc-mg function as a ceRNA. Firstly, overexpression of Inc-mg led to the increased enrichment of Ago2 on Inc-mg but substantially decreased enrichment on *Igf2* (**T-Fig. 1a** above). Secondly, the luciferase activity of *Igf2* 3'UTR reporters was increased upon wild-type

lnc-mg overexpression but not upon miR-125b binding site mutated lnc-mg (**T-Fig. 1b** above). Finally, streptavidin capture analysis suggested that the binding enrichment of miR-125b on *Igf2* decreased with overexpression of wild-type lnc-mg, but not with miR-125b binding site mutated lnc-mg (**T-Fig. 1c** above). Collectively, these new evidences in **T-Fig. 1** above and **Fig. 5** in last version demonstrated that lnc-mg could elevate *Igf2* expression through competing endogenous RNA for miR-125b.

Supplementary Fig. 2d: lnc-mg overexpression is 600-1200 fold of the endogenous. This is a non-physiological level. I suspect it was a PCR artifact - DNA contamination from the transfected lnc-mg-expressing plasmid. The authors need to fix this problem.

A: Thanks for the reviewer's helpful comments. We are very sorry for the confusing data, and we have redetected the over-expression level of lnc-mg in C2C12 cells using the PrimeScript™ RT reagent Kit with gDNA Eraser (Perfect Real Time, Takara). It was found that lnc-mg increased ~20 fold after plasmid DNA erased (**T-Fig. 4**). Because we have removed the C2C12 data in **Supplementary Fig. 2** according to the suggestion "I do not believe the C2C12 data add much information. Unless the authors have a very compelling reason for showing these data in both cell types, I recommend removing the C2C12 data" from reviewer 1, so we showed the data as **T-Fig. 4** below.

T-Figure 4

Real-time PCR analysis of lnc-mg expression in C2C12 cells after transfected with control

vector or Inc-mg vector. Mean values \pm SEM, n=4, * $P < 0.05$.

The authors need to mention the age and sex of the animals used in the experiments. Some genes can have age/sex-specific effects.

A: Thanks for the reviewer's helpful suggestion. In the whole study, we used 8-week old mice, three for male and three for female. We have added the missed information in Figure legends of **Fig. 3** (Please refer to **lines 463-464**), **Fig. 4** (Please refer to **line 478**) and **Fig. 6** (Please refer to **lines 515-516**) in revised manuscript. Because we used half male and half female animals in this study, we have not observed sex-specific effect of Inc-mg on myogenesis. All the mice used in this study are 8 weeks old. Up to now, we have no data about age-specific effect of Inc-mg on myogenesis, and which need to be investigated in the future. Please refer to **lines 463-464**, **line 478**, and **lines 515-516** in updated manuscript.

Response to the comments from Reviewer 3 (NCOMMS-16-16008-T)

General/overarching comments to the author:

This is a comprehensive piece of work, novel and highly interesting. In its present form, the authors need to thoroughly proof read the manuscript. The abstract and introduction, in particular, are littered with inconsistencies. There are also parts of the methods, results, figures and figure legends that require more detail or clarification in order to confirm the studies robustness, as detailed in my specific comments below. The discussion also fails to highlight some of the most important conclusions.

A: Thanks for the reviewer's comments and the suggestions were well taken.

Specific Comments:

Abstract

Authors state: 'Recent studies have highlighted important roles for long noncoding RNAs (lncRNAs) as essential regulators during tissue/organ development.' The authors go on to investigate adult myogenesis in the manuscript, so seems strange to open the abstract to include 'development,' rather than, for example, reference to repair and regeneration of adult skeletal muscle.

A: Thanks for the reviewer's suggestion. We agree with the reviewer that it seems strange using "development" to open the abstract. We have reversed it to "Recent studies have indicated important roles for long noncoding RNAs (lncRNAs) as potential essential regulators of myogenesis and adult skeletal muscle regeneration." Please refer to **lines 54-55** in revised manuscript.

Authors include: 'Recent studies have highlighted important roles for a few long noncoding RNAs (lncRNAs)...' Remove 'a few.'

A: Thanks for the reviewer's suggestion. We have removed 'a few' and revised the sentence to "Recent studies have indicated important roles for long noncoding RNAs (lncRNAs)..." Please refer to **line 54** in revised manuscript.

Authors state: 'explaining the role of lncRNAs in myogenic differentiation of stem cells and myogenesis in vivo.' Revise to: explaining the role of lncRNAs in myogenic differentiation of adult skeletal muscle stem cells (MuSCs) and myogenesis in vivo.

A: Thanks for the reviewer's suggestion. Because of the poor statement and poor structure, so we re-wrote the sentences in abstract with slight modification. We have revised the statement "explaining the role of lncRNAs in myogenic differentiation of stem cells and myogenesis *in vivo*." to "*in vivo*, the role and mechanism of lncRNAs in myogenic differentiation of adult skeletal muscle stem cells (MuSCs) and myogenesis are still largely unknown." Please refer to **lines 55-57** in revised manuscript.

Poor sentence structure, authors write: 'In vivo, skeletal muscle specific knockout of mouse lnc-mg develops muscle atrophy and losing muscular endurance during exercise, while skeletal muscle-specific lnc-mg transgenic mice promotes muscle hypertrophy.'

Consider revising to: In vivo, skeletal muscle specific knockout of lnc-mg resulted in muscle atrophy and the loss of muscular endurance during exercise in mice. Alternatively, skeletal muscle-specific transgenic overexpression of lnc-mg promoted muscle hypertrophy in mice.

A: Thanks for the reviewer's suggestion. We have revised the statement to "*In vivo*, skeletal muscle conditional knockout of lnc-mg resulted in muscle atrophy and the loss of muscular endurance during exercise. Alternatively, skeletal muscle-specific overexpression of lnc-mg promoted muscle hypertrophy in mice." Please refer to **lines 58-61** in revised manuscript.

Authors mention 'In vitro analyses...' this needs expanding to include details of the cells derived. E.g. in-vitro analysis of primary skeletal muscle cells isolated from mice....

A: Thanks for the reviewer's suggestion. We have added the missed information and

revised the statement to “*In vitro* analyses of primary skeletal muscle cells isolated from mice.” Please refer to **lines 61-62** in revised manuscript.

Authors include: 'Showed that Inc-mg is induced during myogenesis.' Details are required here e.g. Inc-mg is increased exponentially over the time course (0-5 days) of myogenesis. Or is increased up to 3 day and then plateaus at 5 days? Reviewer is unaware of the direction and the former sentence is hypothetical as I am still to read the results; however, more detail is required.

A: Thanks for the reviewer’s suggestion. We are sorry for not describing the details about the changing trends and time period of Inc-mg. The data in **Fig. 1d** showed that Inc-mg increased over the time (0-5 days) during MuSCs differentiation. Thus we have revised the statement “showed that Inc-mg was induced during myogenesis.” to “showed that expression of Inc-mg was increased gradually during myogenic differentiation” Please refer to **lines 62-63** in revised manuscript.

Authors state: 'and accelerates muscle stem cells differentiation in vitro.' More detail required e.g. improves muscle cell differentiation in-vitro when overexpressed?

A: Thanks for the reviewer’s suggestion. Base on **Fig. 2b** and **Fig. 2f**, knockdown of Inc-mg resulted in the inhibition of muscle stem cells differentiation and overexpression of Inc-mg accelerated muscle stem cells differentiation *in vitro*. To make it clear, we revised the sentence to “overexpressed Inc-mg improved cell differentiation.” Please refer to **line 63** in revised manuscript.

Author uses terminology: 'Molecular sponge' then goes on to say 'through antagonizing...' I am not sure this terminology holds up within the context of the sentence as a sponge is something that absorbs or perhaps reduces but does not antagonize... The use of antagonize in this context is also not familiar as its definition is to cause (usually someone/something) to become hostile.

A: Thanks for the reviewer’s comment. We agree with the reviewer’s opinion that ‘sponge’

and 'antagonize' are two different terminologies. 'Antagonize' refers to cause (someone or something) to become hostile, whereas 'sponge' is something that absorbs or perhaps reduces something else. Therefore, we have revised the related sentences in updated manuscript.

1. Revised the sentence "lnc-mg acts as a molecular sponge through antagonizing miR-125b to control Igf2 protein level during myogenesis." in last version to "lnc-mg promoted myogenesis, by functioning as a competing endogenous RNA (ceRNA) for miR-125b to control protein abundance of Igf2." Please refer to **lines 63-65** in revised manuscript.

2. Revised the sentence "lnc-mg antagonizes miR-125b *in vivo*." In last version to "lnc-mg modulates miR-125b by functioning as a competing endogenous RNA *in vivo*." Please refer to **line 198** in revised manuscript.

3. Revised the sentence "By acting as a molecular sponge, lnc-mg antagonizes miR-125b to control Igf2 protein level *in vitro* and *in vivo*." in last version to "By functioning as a competing endogenous RNA, lnc-mg blocks miR-125b to control Igf2 protein level *in vitro* and *in vivo*." Please refer to **lines 221-222** in revised manuscript.

4. Revised the sentence "lnc-mg protects Igf2 through antagonizing miR-125b *in vitro*." in last version to "lnc-mg protects Igf2 by functioning as a competing endogenous RNA for miR-125b *in vitro*." Please refer to **lines 480-481** in revised manuscript.

5. Revised the sentence "lnc-mg protects Igf2 through antagonizing miR-125b *in vivo*." In last version to "lnc-mg protects Igf2 by functioning as a competing endogenous RNA for miR-125b *in vivo*." Please refer to **lines 507-508** in revised manuscript.

6. Revised the sentence "Molecular mechanism of lnc-mg antagonizing miR-125b to control Igf2 protein level by acting as a ceRNA (right)." in last version to "Molecular mechanism of lnc-mg: regulating miR-125b to control Igf2 protein level by functioning as a ceRNA." Please refer to **lines 520-522** in revised manuscript.

As opposed to an antagonist in biochemistry that is used to describe an activator of something. Consider revising this sentence. By suggesting, e.g. lnc-mg results in increased miR-125b that subsequently controls protein abundance of Igf2.

A: Thanks for the reviewer's comment. We have revised the sentence "lnc-mg acts as a molecular sponge through antagonizing miR-125b to control Igf2 protein level during myogenesis." in last version to "lnc-mg promoted myogenesis, by functioning as a competing endogenous RNA (ceRNA) for miR-125b to control protein abundance of Igf2." Please refer to **lines 63-65** in revised manuscript.

Authors state: 'miR-125b to control Igf2 protein level during myogenesis...' A little more detail of how it does this and how it was demonstrated in the present study is required.

A: Thanks for the reviewer's comment. Firstly, "lnc-mg acts as a molecular sponge through antagonizing miR-125b to control Igf2 protein level during myogenesis" is an over statement, we have revised it to "lnc-mg promoted myogenesis, by functioning as a competing endogenous RNA (ceRNA) for miR-125b to control protein abundance of Igf2." (Please refer to **lines 63-65**). Secondly, based on our previous *in vitro* and *in vivo* results (refer to **Fig. 5 K-I, Fig. 6**), the WB and Elisa data also showed that the level of Igf2 protein was increased with the over-expression of lnc-mg, while decreased when lnc-mg was knocked down. In addition, we added new supplementary data to support the mechanistic part. Firstly, overexpression of lnc-mg led to the increased enrichment of Ago2 on lnc-mg, while substantially decreased enrichment on *Igf2* (**T-Fig. 1a**). Secondly, the luciferase activity of *Igf2* 3'UTR reporters was increased upon wild-type lnc-mg over-expression but not upon miR-125b binding site mutated lnc-mg (**T-Fig. 1b**). Finally, streptavidin capture analysis suggested that the binding enrichment of miR-125b on *Igf2* decreased with over-expression of wild-type lnc-mg, but not with miR-125b binding site mutated lnc-mg (**T-Fig. c1**). In summary, lnc-mg protected *Igf2* mRNA by competitively binding with miR-125b, resulting in the increased level of Igf2 protein. Please refer to **lines 63-65, Fig. 5 K-I, Fig. 6** in revised manuscript and **T-Fig.1 a, b, c** below.

T-Figure 1

a.

RIP assay of the enrichment of Ago2 on *Inc-mg*, *Igf2* and *Gapdh* relative to IgG in C2C12 cells transfected with *Inc-mg* empty vector (control vector) or *Inc-mg* overexpression vector (*Inc-mg*). Mean values \pm SEM, $n=4$, $*P < 0.05$.

b.

The relative luciferase activity of psiCHECK-2 containing *Igf2* 3'UTR co-transfected with miR-125b, miR-125b and *Inc-mg*, miR-125b and mutated *Inc-mg* respectively. Mean values \pm SEM, $n=4$, $*P < 0.05$.

C.

Streptavidin capture was performed for C2C12 myoblasts co-transfected with bio-miR-NC or bio-miR-125b and *Igf2* 3'UTR, *Igf2* 3'UTR with Inc-mg or *Igf2* 3'UTR with Inc-mg-mut, followed by real-time PCR to detect *Igf2* mRNA and *Gapdh* mRNA levels. Mean values \pm SEM, n=4, * $P < 0.05$.

Authors include: 'development and the mechanisms that how lncRNAs coordinate the thickness of muscle fiber and muscular strength...' This is poorly written consider using: 'mechanisms by which lncRNAs coordinate the size of skeletal muscle fibers and strength.'

A: Thanks for the reviewer's comment. Because of the poor statement and poor structure, so we re-wrote the sentences in abstract with slight modification. We have deleted 'and the mechanisms that how lncRNAs coordinate the thickness of muscle fiber and muscular strength...' in revised manuscript.

Authors state: 'These findings identify Inc-mg as an important noncoding regulator for skeletal muscle.' Consider revising to: These findings identify Inc-mg as a novel and important noncoding regulator for skeletal muscle.

A: Thanks for the reviewer's suggestion. We have revised the sentence to "These findings identify Inc-mg as a novel and important noncoding regulator for muscle cell differentiation and skeletal muscle development." Please refer to **lines 65-66** in revised manuscript.

Introduction

The authors should make sure they clearly distinguish that they are introducing adult myogenesis and not embryonic myogenesis in the opening paragraph. As this is what they go onto present data for in the present study. Reading this, preceding the methods/results sections, makes the reader jump to the results to confirm the model used, so it should be stated up front.

A: Thanks for the reviewer's comments. We re-distinguished adult myogenesis and embryonic myogenesis by referring to literature: during embryonic myogenesis, skeletal muscle was originated from the paraxial mesoderm, which developed into somites before forming dermomyotome, and finally came into myotome. Myoblasts undergo frequent divisions and fuse together leading to the formation of multinucleated myotubes/myofibers. (Buckingham, M. *et al.* 2003). However, myogenesis in adult depend on the activation of satellite cells which have the potential to differentiate into new fibers (Charge, S.B. *et al.* 2004). As reviewer suggested that we were introducing adult myogenesis, thus we revised 'myogenesis' in last version to 'myogenesis in adult'. Please refer to **line 73** in updated manuscript.

Authors state: 'and then proceeds with cell proliferation and fusion.' Important to include migration is this sentence to read: .and then proceeds with cell proliferation, migration and fusion.

A: Thanks for the reviewer's comment. We have revised the sentence to "and then proceeds with cell proliferation, migration and fusion." in the updated manuscript according to the reviewer's suggestion. Please refer to **lines 74-75** in updated manuscript.

Authors include: 'terminal differentiation is initiated to form multinucleated myotubes with contractile ability.' Consider revising to: terminal differentiation is initiated to form multinucleated myotubes with the capacity to contract.

A: Thanks for the reviewer's comment. We have revised the sentence to 'terminal differentiation is initiated to form multinucleated myotubes with the capacity to contract.' Please refer to **lines 75-76** in updated manuscript.

Roles as signaling molecule, should read: Roles as signaling molecules

A: Thanks for the reviewer's suggestion. We have corrected "Roles as signaling molecule" to "Roles as signaling molecules". Please refer to **line 80** in updated manuscript.

Authors state: 'Recently, a large class of lncRNAs, referred to as competing endogenous RNAs (ceRNAs), has been characterized.' This sentence requires a primary reference.

A: Thanks for the reviewer's suggestion. We have added references (Qu, L. *et al.* 2016, Yue, B. *et al.* 2016, Yan, B. *et al.* 2015, Kallen, A.N. *et al.* 2013 and Cesana, M. *et al.* 2011) at the end of sentence "Recently, a class of lncRNAs, referred to as competing endogenous RNAs (ceRNAs), have been characterized." Please refer to **line 94** in updated manuscript.

Authors state: 'ceRNAs protect mRNAs by acting as sponges for microRNAs,' the papers the authors cites refer to these first as 'molecular sponges' so consider revising to: ceRNAs protect mRNAs by acting as 'molecular sponges' for microRNAs.

A: Thanks for the reviewer's suggestion. We have revised the statement to "ceRNAs protect mRNAs by acting as molecular sponges for microRNAs". Please refer to **line 94** in updated manuscript.

Authors state: 'Additionally, H19 has been demonstrated to act as a molecular sponge for let-7 to control skeletal muscle differentiation.' This should be replaced with allowing to read: Additionally, H19 has been demonstrated to act as a molecular sponge allowing let-7 to control skeletal muscle differentiation.

A: Thanks for the reviewer's suggestion. We have revised the sentence to "Additionally, H19 has been demonstrated to act as a molecular sponge regulating let-7 to control skeletal muscle differentiation". Please refer to **lines 99-101** in updated manuscript.

Immediately following the penultimate paragraph (prior to the last paragraph) in the introduction, the authors must to include their aims and specific hypotheses for the study that stems from the extensive rationale for the study the authors include.

A: Thanks for the reviewer's suggestion. We have added the aims and specific hypotheses in the updated manuscript for "While partial functions of these lncRNAs have been identified *in vitro* or even preliminary *in vivo*, most of their roles for myogenesis are still waiting for disclosing. Thus, it is essential to identify functional lncRNAs during myogenesis, and identify its function and mechanism *in vivo*". Please refer **lines 101-104** in updated manuscript.

Authors mention: 'we describe the function of a new lncRNA, named lnc-mg.' How was this name decided? What does 'mg' stand for? This detail is required if this is a newly defined lncRNA. This is also the first time you refer to this in the main text so its full name (not just acronym) should be used here.

A: Thanks for the reviewer's comment. lnc-mg stands for myogenesis-associated lncRNA. 'mg' is originated from its function in promoting muscle stem cell differentiation and myogenesis. We have added the detail in the main text for "In this study, we describe the function of a myogenesis-associated lncRNA (short for lnc-mg) in mice." Please refer to **lines 105-106** in updated manuscript.

Methods

In Supplementary figure 1a the authors use the acroynms GM or DM to describe their growth and differentiation media respectively. However, these acroynms are not specifically described in the methods or figure legend (reviewer suggests to include this information in the methods section, under; 'In vitro cell culture and differentiation' authors should change wording from: 'DMEM with 10% FBS and induced to myogenic differentiation by switching to DMEM containing 2% horse serum.' To: DMEM with 10% FBS (growth media/GM) and induced to myogenic differentiation by switching to DMEM containing 2% horse serum (differentiation media/DM).

A: Thanks for the reviewer's suggestion. We have added the missing information in the methods in the revised manuscript. Please refer to the statement in **lines 243-244**: "DMEM with 10% FBS (growth media/GM) and induced myogenic differentiation by switching the medium to DMEM containing 2% horse serum (differentiation media/DM)." Please refer to **lines 243-244** in updated manuscript.

Under the real-time PCR section author's state in accordance with the MIQE guidelines and state a PMID, Please include a in text citation/reference in line with the journals format.

A: Thanks for the reviewer's suggestion. We added the reference (Bustin, S.A. *et al.* 2013) and (Bustin, S.A. *et al.* 2009) at the end of the sentence "Relative expression values were calculated using the comparative threshold cycle ($\Delta\Delta$ CT) method in accordance with the MIQE guidelines (PMID: 24173381)." Please refer to **line 255** in updated manuscript.

Was there a set volume or concentration of cDNA loaded into the PCR reactions, Please include information.

A: Thanks for the reviewer's suggestion. We have added the missing information about amount of cDNA loaded into the PCR reactions in the methods. Please refer to sentence in **lines 247-249**: "RNA (1 μ g) was reverse-transcribed by using the PrimeScript™ RT reagent Kit with gDNA Eraser (Perfect Real Time) (Takara). One microliter of a 1:5 dilution of the synthesized cDNA was used for real-time PCR analysis."

No detailed information is included on the PCR cycling protocol or number of cycles performed, this is required.

A: Thanks for the reviewer's suggestion. We have added the detailed information of PCR cycling protocol in the methods. Please refer to sentence in **lines 249-252**: "The relative abundance of the mRNAs was determined using SYBR® Premix Ex Taq™ II (Tli RNaseH Plus) (Takara) according to the manufacturer's instructions. The following thermal settings were used: 95°C for 30 s followed by 40 cycles of 95°C for 5 s and 60°C for 30 s."

The authors require more information to ensure appropriate PCR conditions were met. Please include information in the methods on the stability and therefore the suitability of your chosen reference gene (*Gapdh*) across all experimental conditions. E.g. mean \pm Sd and % variation in the reference gene across conditions. If variation was low was a pooled reference ct value used in the delta delta ct equation or was the specific samples reference gene Ct value used.

A: Thanks for the reviewer's suggestion. In order to provide more information to ensure appropriate PCR conditions. Firstly, we analyzed the system stability by testing the Ct value of *Gapdh* for five times. It was found that the Ct value was very similar (**T-Fig. 2a**). It showed that the variation is below 10%, and our data is reliable according to the MIQE. Secondly, we detected the Ct value of *Gapdh* from C2C12 during differentiation and found that the Ct value of *Gapdh* was relatively stable during differentiation (**T-Fig. 2b**). According to MIQE, if a certain gene is expressed stably in both undifferentiated and differentiated cells, this gene could be employed as a reference gene. Thus, we chose *Gapdh* as reference gene in our research. Please refer to **T-Fig. 2a, b** below.

T-Figure 2

a.

The Ct value of *Gapdh* detected by real-time PCR for five times. Mean values \pm SEM, n=4.

b.

The Ct value of *Gapdh* detected by real-time PCR during induced differentiation of C2C12 cells. Mean values \pm SEM, n=4.

How did you confirm that your PCR was specific to the gene of interest e.g. did you run at least a melt curve analysis that suggested a single product for each gene investigated.

A: Thanks for the reviewer's comment. We have run the melting curve of all the primers used in our study to confirm their specificity. Please refer **T-Figure 3** below.

T-Figure 3

The melt curves of all the primers in this study

For example, upon performing BLAST searches for your myogenin primers taken from Macpherson et al., (Cell Biochem 2011), they amplify a product of 178 bp on the myogenin gene. However, it also shows potential unintended target of the Fem1b gene of 331bp, although a little large for real time PCR, a melt analysis would at least confirm that one single product was amplified.

A: Thanks for the reviewer's comment. Considering the reviewer's suggestion, we selected another pair of myogenin primers that was reported by Y. Oikawa, Y. *et al.* 2011.

It was suggested that the primers were specific according to the results of BLAST and melting curve (Please refer to **T-Fig. 4a** below). Meanwhile, the expression change of myogenin is similar to the previous results (Please refer to **T-Fig. 4b** below).

T-Figure 4

a.

The BLAST search figure and melting curve of *myogenin* primers in revised manuscript.

b.

Real-time PCR analysis of *MyoG* expression in C2C12 cells transfected with control vector or Inc-mg expression vector by old and new *myoG* primers. Mean values \pm SEM, $n=4$, $*P < 0.05$.

Please include mean, SD and variation for the efficiency of PCR reactions across all conditions. If there was large variation e.g. above 10% was the analysis tailored to account for this e.g. use of REST software etc.

A: Thanks for the reviewer's comment. As the reviewer suggested, we re-analyzed the system error for the efficiency of PCR reactions by detecting Inc-mg, *MyoD*, *MyoG*, *miR-125b*, *Gapdh* and *U6* in C2C12 cells for three times. It was found that the variation

was below 10% (T-Figure 5 below). Thus, we did not use REST software in our study.

T-Figure 5

The Ct value of all the primers in this study detected by real-time PCR for three times. Mean values \pm SEM, n=4.

What was used as the calibrator condition in the relative delta PCR analysis. Include this in the methods or in the figure legends where the PCR data is included.
A: Thanks for the reviewer's suggestion. We have added the calibrator condition in 'y-axis' of Fig. 1b-d, Supplementary Fig. 1b, c, Fig. 2a, d, e, h, Fig. 5b, e-j, Supplementary Fig. 3c and Supplementary Fig. 4 in revised manuscript".

In table1- PCR primers, Please include the product lengths (bp) for each gene e.g. a blast of your myoD primers, these amplify a product of 121bp. Myogenin 178bp etc. In table 1 Please include the accession number for each of the genes e.g. myoD is NM_010866.2.

A: Thanks for the reviewer's suggestion. We have added the product lengths (bp) and the accession number for each of the genes in **Supplementary Table 2.**

Cell Transfection section:

When the author uses a weight of e.g. Inc-mg expression plasmid (500 ng). Is this the final total amount in each 24 well plate or is this 500ng/ml of media with 0.5 or 1 ml of media used per well? Please make this clear.

A: Thanks for the reviewer's comment. 500 ng is the final total amount of Inc-mg expression plasmid in each 24 well plate. The media of each well is 0.5 ml. To make it clear, we have added the information in the method of cell transfection section in the updated manuscript. Please refer to **line 259, line 260, line 262** in revised manuscript.

Also under 'cell transfection', you need to include a reference to the method of confirming Inc-mg silencing/overexpression. Expect this was PCR but authors need to state this, what they did and perhaps refer to the specific figure legends where the silencing/overexpression are demonstrated.

A: Thanks for the reviewer's suggestion. It was reported that MuSCs were transfected successfully by Lipofectamine 3000 reagent in the research of Zhao, Q. *et al.* 2016. Thus we used Lipofectamine 3000 reagent and detected the transfection efficiency by transfecting Cy3-siRNA into MuSCs (**T-Figure 6** below). It was found that the transfection efficiency of Lipofectamine 3000 reagent is acceptable for further experiments. To make it clear, we have added the methods of transfection in **lines 449-450**: "Transient transfection of MuSCs with control shRNA or Inc-mg shRNA, control vector or Inc-mg vector by using Lipofectamine 3000 reagent." and **lines 504-505**: "Transient transfection of C2C12 cells using Lipofectamine 3000 reagent." Please refer to **T-Figure 6** below and **lines 449-450** and **lines 504-505** in revised manuscript.

T-Figure 6

Transfection of Cy3-siRNA into MuSCs by Lipofectamine 3000 reagent after 36 h. Scale bar: 40 μ m.

Under 'immunofluorescent staining': Your antibody for MyHC detects only adult type IIx myHCs. If staining primary cells while fusing or when differentiated you need to justify why you used this particular myHC at the time points the cells/myotubes were fixed and stained for myHC IIx.

A: Thanks for the reviewer's comment. In the publication of Maggs, A.M. *et al.* 2000, MyHC antibody A4.1025 was shown to recognize all types of myHC. From the results of **Fig. 2b, f** in our last version, MyHC antibody A4.1025 was used to indicate the differentiation degree of primary cells. Here, we also redid the MyHC staining experiments using the antibody-MF20 (Developmental Studies Hybridoma Bank, University of Iowa) which is widely used to stain myotubes (Zan, X.J. *et al.* 2013.). Please refer to **Fig. 2b, f** in revised manuscript.

Under. 'Exercise performance test': Authors state: Briefly, mice were first accustomed to treadmill running on a 20{degree sign} incline and 25 cm/s belt speed for 3 days. Did the mice run on treadmills for 3 days without rest? How long did they exercise per day over the three day period. Was this 20 minutes like on the 4th day of exercise as specified in the next sentence?

A: Thanks for the reviewer's comment. To make the method clear, we added the details in the revised method. Please refer to **lines 335-341** list as below: 'In the first day, 10 min running in the morning and 10 min running in the afternoon was employed at 20 cm/s belt speed without incline; In the second day, 10° incline 20 cm/s belt speed in the morning and 10° incline 25 cm/s in the afternoon was used for each 15 min running; In the third day, 25° incline 25 cm/s belt speed in the morning and 25° incline 30 cm/s in the afternoon was put into use for each 20 min running; In the fourth day, mice ran on a 25° incline and 30 cm/s belts peed for 20 min, and then the belt speed was increased by 4 cm/s every 20 min until the mice were exhausted'. Please refer to **lines 335-341** in revised manuscript.

Under section Anti-Ago2 immunoprecipitation, author requires a SPACE after 25 mM Tris-HCl.

A: Thanks for the reviewer's comments. We have added a SPACE after 25 mM Tris-HCl in the updated manuscript. Please refer to **line 387** in revised manuscript.

Also, where author's state: 'magnetic beads was added' should read, magnetic beads were added.

A: According to the reviewer's comment, we have corrected the sentence. We have revised "magnetic beads was added" to "magnetic beads were added". Please refer to **line 390** in revised manuscript.

At the end of this section the author's state: The RNA was extracted from the remaining beads with TRIzol Reagent (Life Technologies) and evaluated by real-time PCR assay. What exactly was evaluated by PCR and what was the specific protocol, if different or if the same as previous PCR methods then refer back to the methods above.

A: Thanks for the reviewer's comments. We assessed the interaction between lnc-mg and miR-125b by testing the amount of lnc-mg, miR-125b and *Gapdh* in RNA extracted from the remaining beads. Except for the primers used for reverse transcription, the protocol for real-time PCR is the same to previous PCR methods. To make it clear, we added the PCR methods in section of "Anti-Ago2 immunoprecipitation". Please refer to sentence of **lines 392-394**: "The RNA was extracted from the remaining beads with TRIzol Reagent (Life Technologies) and evaluated by real-time PCR assay, which is the same to the previous real-time PCR protocol". Please refer to **lines 392-394** in revised manuscript.

Under section: 'Biotin-labeled miR-125b capture,' the final sentence you mention, once biotin labeled, RNA was extracted and PCR assay performed. What were specific PCR requirements for this particular analysis, information is required.

A: Thanks for the reviewer's comments. In section "Biotin-labeled miR-125b capture", the protocol for real-time PCR is the same as previous PCR method for Anti-Ago2

immunoprecipitation. To make it clear, we added the missing information in **lines 406-407** for “The entire PCR assay is the same to the previous real-time PCR protocol”. Please refer to **lines 406-407** in revised manuscript.

The authors also perform microarrays for miRNA's prior to the above reporter assay. There is no detail of these microarray methods in the methods text.

A: Thanks for the reviewer’s suggestion. We have added the missing information in the revised methods. “The microarray experiments were performed by RiboBio (Guangzhou, China). Briefly, total RNAs was extracted from cells of Inc-mg overexpression or Inc-mg knockdown by TRIzol Reagent (Life Technologies). Then the RNA quality was assessed by formaldehyde agarose gel electrophoresis, quantified spectrophotometrically and Agilent 2200 Bioanalyzer (Agilent, USA). 1.5 ug total RNA was labelled on Cy3 using ULS. Then the CustomArray™ microarray was pre-hybridized in nuclease-free water at 65°C for 10 min, and then loaded the microarray onto the rotisserie in the hybridization oven and incubated at 37°C for 60 min with gentle rotation. The hybridization solution was prepared with labeled microRNA target and denatured the hybridization solution at 95°C for 3 min, and then cooled for 20 s on ice. Then the microarray was loaded with the hybridization solution and incubated at 37°C for 16 h with gentle rotation. Then, we removed the microarray from the hybridization solution and washed the microarray by using the wash solution to remove nonspecific hybridization. Then covered the semiconductor microarray surface with the imaging solution and loaded the microarray into the scanner to scan. The data was analyzed by Guangzhou Ribo Bio Co., Ltd.” Please refer to **lines 343-356** in revised manuscript.

There is nothing in the methods as to how the authors measured muscle weight and when (at what time point) animals were sacrificed for this.

A: Thanks for the reviewer’s suggestion. We added the methods of measuring muscle weight and time points of sacrificing animals. Please refer to sentence in **lines 319-321**: “The SOL, EDL, GAS and TA of 8-week-old (three for male and three for female)

Inc-mg^{Skl-/-}, WT and TG mice were harvested and weighed by electronic scale with the minimum range for 1/10000 g." Please refer to **lines 319-321** in revised manuscript.

Also, measurement of force and specific force of the tissue is not detailed in the methods.

A: Thanks for the reviewer's suggestion. Mice were anesthetized via intra-peritoneal injection of a cocktail containing 25 mg/ml ketamine, 2.5 mg/ml xylazine and 0.5 mg/ml acepromazine at 2.5 ml/g body weight. The entire gastrocnemius was isolated and preserved in Ringer's solution which was continuously aerated with 95% O₂ and 5% CO₂ and maintained at 37°C. The distal end of gastrocnemius was connected to an isometric transducer. The gastrocnemius was stimulated with electrical stimulation of 100 Hz. Optimal muscle length was multiplied by 0.85 to calculate the optimal fiber length. Tetanic contractions induced by optical and electrical stimulation were 2 s long with an interval of 3 min between stimulations. Maximal force (M) was analyzed for single twitch contractions. Specific force was normalized for muscle CSA which was calculated by mass (mg)/fiber length (mm)*1.06 (mg/mm³). The specific force was calculated by M/CSA. To make it clear, we add the missed information in revised method. Please refer to **lines 321-331** in revised manuscript.

Histological analysis of muscle sections stained with hematoxylin and eosin (Figure 3b) is also not detailed. This information is required in the methods.

A: Thanks for the reviewer's suggestion. We added the method of H&E in the revised text. The gastrocnemius were fixed in 4% paraformaldehyde, processed and embedded in paraffin prior to sectioning (10 microns) and staining. The tissues were placed in 1% osmium tetroxide in 0.1% M cacodylate buffer for 1 h at 4°C. Then dehydrated and embedded in a pure epoxy resin which became solid after 48 h at 60°C. Then made semithin sections and eluted the epoxy resin, and stained with hematoxylin and eosin (H&E). Reference: M. Lagouge et al., Resveratrol improves mitochondrial function and protects against metabolic disease by activating SIRT1 and PGC-1alpha. Cell 127, 1109-1122 (2006). Please refer to **lines 312-317** in revised manuscript.

ELISA analysis, what were the excite/emission spectra you detected at.

A: Thanks for the reviewer's comment. The excite/emission spectra was detected at 450 nm. To make it clear, we added missed information in method. Please refer to **line 424** in revised manuscript.

Results

Authors state in the results text under the, 'Inc-mg is induced in myogenesis' section: 'Microarray data from quiescent and' how do the authors know these were quiescent versus actively cycling. In Fig 1a, the microarray data shows the panel on the left as GM, so these cells were in growth media 10% serum. A typical skeletal muscle cell quiescent media contains only 0.1% serum. Do the authors simply mean undifferentiated muscle cells?

A: Thanks for the reviewer's comment. We are sorry for mis-describing the cells. Here, the cells used in our study include both undifferentiated and differentiated MuSCs. We have revised the statement to "Microarray data from undifferentiated and differentiated MuSCs revealed that 70 lncRNAs were upregulated and 12 were downregulated during cell differentiation (Fig. 1a and Supplementary Table 1)". Please refer to **lines 115-117** in revised manuscript.

As suggested above, there is no text in the methods for microarray protocols.

A: Thanks for the reviewer's suggestion. We have added the missing information in the revised methods. "The microarray experiments were performed by RiboBio (Guangzhou, China). Briefly, total RNAs was extracted from cells of Inc-mg overexpression or Inc-mg knockdown by TRIzol Reagent (Life Technologies). Then the RNA quality was assessed by formaldehyde agarose gel electrophoresis, quantified spectrophotometrically and Agilent 2200 Bioanalyzer (Agilent, USA). 1.5 ug total RNA was labelled on Cy3 using ULS. Then the CustomArray™ microarray was pre-hybridized in nuclease-free water at 65°C for 10 min, and then loaded the microarray onto the rotisserie in the hybridization oven and incubated at 37°C for 60 min with gentle rotation. The hybridization solution was

prepared with labeled microRNA target and denatured the hybridization solution at 95°C for 3 min, and then cooled for 20 s on ice. Then the microarray was loaded with the hybridization solution and incubated at 37°C for 16 h with gentle rotation. Then, we removed the microarray from the hybridization solution and washed the microarray by using the wash solution to remove nonspecific hybridization. Then covered the semiconductor microarray surface with the imaging solution and loaded the microarray into the scanner to scan. The data was analyzed by Guangzhou RiboBio Co., Ltd.” Please refer to **lines 343-356** in revised manuscript.

Author's state under 'Inc-mg is induced in myogenesis'. and enriched in skeletal muscle.' This should read that was enriched in skeletal muscle.

A: Thanks for the reviewer’s suggestion. We have revised this sentence to ‘which was enriched in skeletal muscle.’ Please refer to **line 118** in revised manuscript.

The authors briefly refer to figure 1c, but the text describes fig1b and there is no description of the data in fig1c on the expression of Inc-mg between muscle groups that show some quite different expression values between muscles.

A: Thanks for the reviewer’s comment. We have described the data in revised text for “In order to validate whether Inc-mg differently expressed in various types of muscles, we examined the levels of Inc-mg in different types of muscles. It was found that expression levels of Inc-mg have only a little difference among different types of muscles, while is higher than other tissues (Fig. 1c and Supplementary Fig. 1c).” Please refer to **lines 119-122** in revised manuscript.

In supplementary figure 1, figure legend the authors state: 'Characterization of Inc-mg. (a) MyHC immunostaining of undifferentiated muscle stem cells (DM) and 5 days of differentiated muscle stem cells (GM).' This does not make complete sense as GM is used for cell proliferation not differentiation and DM media is used for differentiation and should not therefore be linked with undifferentiated muscle stem

cells as the description suggests. I think the DM and GM in brackets need swapping around.

A: Thanks for the reviewer's comment. We are sorry for misused the words DM and GM. We have revised them in revised manuscript. Please refer to **line 525** in revised manuscript.

In the figure legend for figure 1. Authors state: '(skeletal muscle value was normalized to 1.0).' As suggested in the reviewer's methods comment above the appropriate calibrator condition in the relative gene expression analysis should be referred to here.

A: Thanks for the reviewer's comment. Similar to the answer in the method above, the calibrator condition of **Fig. 1b** in last version is: we set the value of Inc-mg in skeletal muscle as 1.0, and then normalized the expression levels of Inc-mg in other tissues to skeletal muscle. However, similar to the reply for reviewer 1, "To show the results in a more reliable way and to highlight the elevated expression of Inc-mg in skeletal muscle, we provided the fold changed data by real-time PCR in revised **Fig. 1b** and added the calibrator condition in 'y-axis' of **Fig. 1b**. Please refer to **Fig. 1b** in revised manuscript.

Furthermore, in the authors supplementary figure 1 the authors state: 'muscle stem cells were treated with Calf Intestine Alkaline Phosphatase (CIP) to remove free 5'-P, then treated with Tobacco Acid Pyrophosphatase (TAP) to remove the cap structure and then a RNA adapter oligonucleotide was ligated to the RNA population using T4 RNA ligase (FistChoice RLM-RACE Kit, Ambion). (f) Bioinformatics analysis of the coding capability of Myh1, H19 and Inc-mg.' These methodologies are not included in the method sections. I am not sure what journal guidelines say about data shown in supplementary figures, but I still think all the methods should be present in the main methods section as the sup figure 1 is referred to in the main results text.

A: Thanks for the reviewer's suggestion. We have added the methods of judging whether Inc-mg has 5' cap or 3' poly A and calculating the coding capability in method sections in the updated manuscript. Please refer to new-added methods below: "The RNA sequences

of *Myh1*, *H19* and *Inc-mg* were put into the Coding Potential Calculator (CPC) program, and both *H19* and *Inc-mg* were predicted to be non-coding RNAs, while *Myh1* was identified to code for protein in method sections. For 3' poly A detection, total RNAs was extracted from muscle stem cells in TRIzol Reagent (Life Technologies) according to the manufacturer's instructions. Then the RNA quality was assessed by formaldehyde agarose gel electrophoresis, quantified spectrophotometrically and Agilent 2200 Bioanalyzer (Agilent, USA). Ribosomal RNA was removed using the Ribo-Zero Magnetic Kits (Illumina) according to the manufacturer's instructions. Then polyA+ RNA fraction and polyA- RNA fraction were isolated by using NEBNext[®] Poly (A) mRNA Magnetic Isolation Module (NEB, NEB #E7490S/L). And the amount of *Inc-mg* was examined in PCR assay with polyA+ RNA fraction and polyA- RNA fraction respectively. For 5'cap detection, the experiment was performed using the FirstChoice RLM-RACE Kit (Ambion) according to the manufacturer's instructions. In brief, total RNAs from muscle stem cells was treated with Calf Intestine Alkaline Phosphatase (CIP) to remove free 5'-P then treated with Tobacco Acid Pyrophosphatase (TAP) to remove the cap structure and then a RNA adapter oligonucleotide was ligated to the RNA population using T4 RNA ligase. The reverse transcribed was performed using primers corresponding to the 5' RACE Adapter sequence provided with the system. PCR amplification was then performed using Taq DNA polymerase (Takara). Please refer to **lines 358-376** in revised manuscript.

Phase/contrast images in figure 2 are really poor quality; the cells/myotubes can hardly be distinguished. The authors should replace with better quality images or remove them and simply keep the fluorescent images only.

A: Thanks for the reviewer's comments and the suggestions were well taken. We redo the MyHC staining experiments and take photos with better quality. We displayed the phase photo as **T-Figure 7** and replaced fluorescence image with better quality in revised **Fig. 2b** and **Fig. 2f**. Please refer to **Fig. 2b** and **Fig. 2f** in revised manuscript and **T-Figure 7** below.

T-Figure 7

MyHC immunostaining of MuSCs transfected with control shRNA or Inc-mg shRNA then cultured in DM for five days. Scale bar: 40 μ m.

MyHC immunostaining of MuSCs transfected with control vector or Inc-mg vector then cultured in DM for five days. Scale bar: 40 μ m.

In the final sentence in section 'Inc-mg promotes MuSCs differentiation', authors need to remove extra space and replace comma where it reads: In addition, Inc-mg

A: Thanks for the reviewer's suggestion. We have corrected the statement to "In addition, Inc-mg". However, for we removed the C2C12 data in **Supplementary Fig. 2** at this version as the reviewer 1 suggested, thus we didn't show the sentence in revised manuscript.

The reviewer thinks that supplementary figure 2 should not be a supplementary file but a figure for the main manuscript, as this is key for the generation of the following/later data.

A: Thanks for the reviewer's suggestion. As the reviewer 1 suggested "I believe that general data showing changes using primary muscle cells is much more relevant and valuable than C2C12 data. In addition, in this case, the data in the two cell types looks exactly the same; therefore I do not believe the C2C12 data add much information. Unless the authors have a very compelling reason for showing these data in both cell types, I recommend removing the C2C12 data". Thus we removed the C2C12 data in Supplementary Fig. 2 at this version.

Under, Conditional knockdown of Inc-mg in skeletal muscle mice results in muscle atrophy and weakness in vivo, authors state:' Moreover, the muscle fiber in gastrocnemius cross sections was thinner in Inc-mg^{ski-/-} mice (Fig. 3b, c). Measurement of the mean diameter of muscle fibers revealed that Inc-mg^{ski-/-} mice had more thin fibers (Fig. 3d).' Again there is no reference to how these parameters were measured in the methods section.' Also, this is poorly written consider revising to: **Moreover, the cross-sectional area of muscle fibers in the gastrocnemius were smaller in Inc-mg^{ski-/-} mice (Fig. 3b, c). Measurement of the mean diameter of muscle fibers revealed that Inc-mg^{ski-/-} mice had a larger number of thinner fibers (Fig. 3d).**

A: Thanks for the reviewer's suggestion. We have added the methods of measuring the cross-section area in the methods section. Please refer to the revised method (**lines**

303-310): The measurement of cross-section area was performed according to the published method (Deng. B. *et al.* 2012) with slight modification. The cross-section area of each muscle (Fibers number, $n=500$) in four fields from each animal of six mice (8-week old C57BL/6J mice, three for male, three for female) were randomly chosen and determined using the ImageJ program, and then calculated the mean cross-section area of each group. Fiber diameter was calculated as the caliper width perpendicular to the longest chord of each fiber. The total fiber number was calculated using an image of 20× magnification from the entire field of muscle section which was randomly chosen.

We have revised the sentence to “Moreover, the cross-sectional area of muscle fibers in the gastrocnemius was smaller in *Inc-mg^{ski-/-}* mice (Fig. 3b, c). Measurement of the mean diameter of muscle fibers revealed that *Inc-mg^{ski-/-}* mice had a larger number of thinner fibers (Fig. 3d).” (Please refer to **lines 146-148**). Please refer to **lines 303-310 and lines 146-148** in revised manuscript.

There is no mention of specific force in the results text (but you include a figure under fig 3e.) in the results text and the differences in control vs *Inc-mg^{ski-/-}*.

A: Thanks for the reviewer’s comment. As the reviewer suggested, we have revised this sentence to “the force and specific tetanic force (Fig. 3e), as well as muscle performance in forced treadmill running tests (Fig. 3f) were all reduced in *Inc-mg^{ski-/-}* mice.” Please refer to **lines 148-150** in the updated manuscript.

Figure 3: Why are the CSA's relative to control and not absolute e.g. mm²

A: Thanks for the reviewer’s comment. In order to assess the fold change of CSA between gene-knockdown group and WT group, we set the value of CSA in control group as 1.0, and then normalized the gene-knockdown group to the control. We have added the information in ‘y-axis’ of **Fig. 3b**. Please refer to **Fig. 3b** in revised manuscript.

Figure 3: (d) Distribution and mean diameter of muscle fibers in gastrocnemius. How is the y axis for 'ratio' calculated, why is this not simply a frequency or number of fibers that fall within each category of fiber diameter.

A: Thanks for the reviewer's comment. This is a method which we referred to the research of Sartori. R. *et al.* 2013. That it is, "fiber diameter was calculated as the caliper width perpendicular to the longest chord of each myofiber". The 'y axis' for 'ratio' in **Fig. 3d** refers to 'mean number of each category of fiber diameter /total number'. We have showed the information in **Fig. 3d** in revised manuscript. Please refer to **Fig. 3d** in revised manuscript.

Figure 4g. Does denervation induce a loss of muscle in the control group as you would expect. I.e. are the white bar controls vs. denervated in WT significantly different. If they were not, then authors need to explain why this is the case.

A: Thanks for the reviewer's comment. As we speculated, denervation induced a loss of muscle compared to the WT mice (Please refer to **T-Figure 8** below). In addition, the CSA was significantly decreased in denervated mice in WT group (**Fig. 4g**). Therefore, it was showed that the denervated model is successful in our study. Please refer to **Fig. 4g** in revised manuscript and **T-Figure 8** below.

T-Figure 8

The CSA in denervated mouse is significantly decreased in WT mice. Scale bar: 50 μ m

In figure 4g. What are the percent changes in control WT vs. TG and denervated WT vs. TG. Are they comparable or different, if different are they significantly different? It looks like control increases from 100% in WT to 160% in TG (approx. 60% increase) and in the denervated groups CSA increases from 75% in WT to 125% in TG (approx. a 50% increase). Therefore, as questioned above, is this increase

significantly different or not? E.g. it is quite extraordinary that on a background of denervation that the response/magnitude of increased muscle size to overexpression of Inc-mg is similar (50-60%) to that of WT. Authors should discuss this in more detail in the discussion as this is an intriguing finding and relevant for age-related muscle loss where we know denervation occurs.

A: Thanks for the reviewer's comment. As seen in **Fig. 4g**, CSA in TG mice is about 60% higher compared to WT mice. After denervation, CSA decreased about 25% in WT mice and decreased 20% in TG mice. In the TG mice, due to Inc-mg over-expression, the muscle can grow significantly bigger and get a much higher peak muscle mass than the WT mice. Thus, when muscle was denervated, the reduction of muscle mass in the TG is not as obvious as that in WT mice. However, we found that the reduction ratio (WT 25% vs TG 20%) with no significant difference. Then, we detected the expression of Inc-mg in WT mice after denervation. Unexpectedly, it was found that Inc-mg expression is not significantly changed after denervation in muscle (**T-Figure 9** below). Thus, we speculated that Inc-mg is not involved in denervation induced muscle loss, which need to be carefully investigated in the future. To make it clear, we added the discussion information about this interesting finding in **lines 222-225**). Please refer to **fig. 4g** and **lines 222-225** in revised manuscript and **T-Figure 9** below.

Real-time PCR analysis of Inc-mg expression after denervation in WT mice. NS= no statistic difference. Mean values \pm SEM, n=4, * $P < 0.05$.

Figure 6 includes real time pcr analysis of miR125b. What are the primer/probe details and specific pcr requirements.

A: Thanks for the reviewer's suggestion. We have added the real-time PCR primer sequence of miR-125b in **Supplementary Data 2 miR-125b**

RT primer: GTCGTATCCAGTGCAGGGTCCGAGGTATTTCGCACTGGATACGACTCACAA;

miR-125b-Forward: CACGCATCCCTGAGACCC;

miR-125b-Reverse: CCAGTGCAGGGTCCGAGGTA

In addition, similar to the reviewer's question in methods, except for the primers for reverse transcription, the protocol for real-time PCR is the same to previous PCR method.

Discussion

First sentence: 'Function as sponges,' revise to, function as molecular sponges.

A: Thanks for the reviewer's comment. We have revised the sentence to "function as molecular sponges." in the updated manuscript. Please refer to **line 209** in revised manuscript.

Authors state: 'resulting in increased Igf2 protein to enhance myogenesis.' Authors should include original work on Igf2 improving differentiation e.g. Stewart et al., 1996 PMID: 8841419 and Stewart and Rotwein 1996, PMID: 8626686

A: We appreciate very much for the references recommended by the reviewer. We have added these references at the end of Igf2 in the discussion. Please refer to **line 229** in revised manuscript.

Authors require a comma after: 'and mechanistic characterization of lnc-mg, both...'

A: Thanks for the reviewer's comment. We have revised the sentence to "and mechanistic characterization of lnc-mg, with both..." in the updated manuscript. Please refer to **lines 219-220** in revised manuscript.

The final paragraph in the discussion: 'Although lnc-mg homolog in humans has not yet been found, we cloned potential homologous lncRNAs from pig and sheep.

Because skeletal muscle-specific Inc-mg transgenic mice had much bigger and stronger muscle compared to their littermates, we further generated muscle-specific Inc-mg homologous IncRNA transgenic sheep. These sheep were much bigger with enhanced muscularity compared to wild type lambs (data not shown). Consequently, in addition to providing an interesting new mechanism that regulates myogenesis in several mammalian species, Inc-mg and its homologous genes could serve as useful candidates to improve muscle production in farm animals.' This is highly speculative and not a conclusion based on the data presented. Unless authors plan to include the lamb data they refer to then conclusions should be made within the context of the data presented.

A: Thanks for the reviewer's suggestion. Consider that the transgenic sheep study is still ongoing, we have removed this paragraph in revised manuscript.

The authors should focus on the interesting findings (as suggested in the results comment above) on the rescue of muscle size following denervation in transgenic animals for Inc-mg as a key finding.

A: Thanks for the reviewer's comment. As seen in **Fig. 4g**: in TG mice, due to Inc-mg overexpression, the muscle grew significantly bigger and get a higher peak muscle mass than WT mice. Thus, when muscle was denervated, the reduction of muscle mass in the TG is smaller than that in WT mice. In addition, the reduction ratio (WT 25% vs TG 20%) with no significant difference. We speculated that Inc-mg is not involved in denervation induced muscle loss, because Inc-mg expression is not significant changed after denervation in WT mice muscle (**T-Figure 10** below). However, we think that the key finding in this study is the effect and mechanism of Inc-mg to promote myogenesis but not the rescue of muscle size following denervation in transgenic mice. Please refer to **Fig. 4g** in revised manuscript and **T-Figure 10** below.

T-Figure 10

Real-time PCR analysis of Inc-mg expression after denervation in WT mice. NS= no statistic difference. Mean values \pm SEM, n=4, * $P < 0.05$.

REVIEWERS' COMMENTS:

Reviewer #1 (Remarks to the Author):

The authors have address all my comments.

Reviewer #2 (Remarks to the Author):

The authors have successfully addressed all my concerns with new data.

Reviewer #3 (Remarks to the Author):

The authors have responded to all my original comments appropriately.

I still do not think the new aims/hypothesis are appropriate, as the amended sentence still does not include an aim or hypothesis, rather more rationale. This should be amended prior to acceptance.